# A Multivariate Time Series Analysis of Ground Deformation Using Persistent Scatterer Interferometry

Serena Rigamonti *, Giuseppe Dattola, Paolo Frattini and Giovanni Battista Crosta

Department of Earth and Environmental Sciences (DISAT), University of Milano-Bicocca, 20126 Milano, Italy;
giuseppe.dattola@unimib.it (G.D.); paolo.frattini@unimib.it (P.F.); giovannibattista.crosta@unimib.it (G.B.C.)
* Correspondence: s.rigamonti12@campus.unimib.it

**Abstract:** Ground deformations in urban areas can be the result of a combination of multiple factors and pose several hazards to infrastructures and human lives. In order to monitor these phenomena, Interferometric Synthetic Aperture Radar (InSAR) techniques are applied. The obtained signals record the overlapping of the phenomena, and their separation is a relevant issue. In this framework, we explored a new multi-method approach based on the combination of Principal Component Analysis (PCA), Independent Component Analysis (ICA) and Hierarchal Clustering (HC) on the standardized results to distinguish the main trends and seasonal signals embedded in the time series of ground displacements, to understand spatial-temporal patterns, to correlate ground deformation phenomena with geological and anthropogenic factors, and to recognize the specific footprints of different ground deformation phenomena. This method allows us to classify the ground deformations at the site scale in the metropolitan area of Naples, which is affected by uplift cycles, subsidence, cavity instabilities and sinkholes. At the local scale, the results allow a kinematic classification using the extracted components and considering the effect of the radius of influence generated by each cavity, as it is performed from a theoretical point of view when the draw angle is considered. According to the results, among the classified cavities, 2% were assigned to subsidence and 11% to uplift kinematics, while the remaining were found to be stable. Furthermore, our results show that the centering of the Spatial-PCA (S-PCA) is representative of the region's main trend, whereas Temporal-PCA (T-PCA) gives information about the displacement rates identified by each component.

**Keywords:** InSAR; Ground deformation; Subsidence; PCA; ICA; Hierarchical Clustering; Kinematic cavity classification

## 1. Introduction

Ground deformations in urban areas can pose several hazards to structures, infrastructures and human lives. For this reason, the areas affected by ground deformations require appropriate monitoring systems, analyses, and methodologies to implement the necessary risk mitigation strategies. In this context, the application of interferometric synthetic aperture radars (InSAR) for measuring and monitoring ground deformations has become a growing practice in recent years, thanks to the development of Persistent Scatterer Interferometry (PSI) [1]. PSI makes it possible to detect and monitor the displacement of anthropic or natural elements whose radar signature remains consistent over long time periods. Hence, this technique enables us to analyze the spatiotemporal evolution of possible deformation phenomena of the Earth's surface, along the satellite Line-Of-Sight (LOS), providing displacement time series over a sparse network of radar targets. With the exception of heavily vegetated areas, forests and agricultural fields, the spatial resolution of measurement points is much higher than what is typically available with ground-based geodetic instruments, in particular over urban areas.

Ground deformations often result from complex superimpositions of natural and human factors acting simultaneously at different temporal and spatial scales. For instance,

in urban areas, several factors may have an impact on ground deformations, overloading and structural adjustment, underground excavations, groundwater drainage or over-exploitation, consolidation of fine sediments and soil rich in organic matter superimposing to active tectonics, and volcano dynamics [2–6].

To separate these overlapping phenomena, different techniques based on the time-series analysis have been proposed. Among them, Principal Component Analysis (PCA) and Independent Component Analysis (ICA) have been employed [7,8]. Recent studies applied PCA on InSAR datasets to identify land subsidence and uplift related to Phlegraean Fields [9], uplift in urban areas induced by multiple factors [2,6], and landslides [10]. For instance, PCA was applied by [11–13] considering meteorological variables, whereas [14] applied T-PCA to isolate temporally variable deformation patterns embedded in multi-decadal time series. Finally, ref. [3] applied PCA to filter data and correct artefacts introduced by measurement procedures. A combination of PCA and clustering was also used to classify regions by grouping the results [12,13].

In a similar way to PCA, ICA has been widely applied to the prediction and extraction of signals in multivariate time series. For example, ref. [15] compared the performance of three different ICA algorithms ([16,17]). Refs. [18–21] applied ICA to GPS coordinate time series to separate seasonal signals, to analyze the spatial and temporal characteristics of land subsidence and to characterize the background level of deformation and volcanic signals in the presence of atmospheric noise. Ref. [22] tested a modified variational Bayesian ICA method (vbICA) to recover multiple sources of ground deformation even in the presence of missing data. Some authors, such as [23], applied ICA, PCA, and also functional curve fitting (FCF) to InSAR time series; through PCA and ICA, they separated independent spatial-temporal components of the deformation, illustrating different geomechanical processes on and around the studied landslide. In particular, these authors identified the mechanisms by means of PCA and they improved their results by ICA optimization. This improvement was also observed in our study, in which the same regions obtained by PCA were also identified by ICA, but the latter is more effective at separating monotonic trends from the seasonal signals.

The metropolitan area of Naples is affected by numerous ground deformation phenomena such as bradyseism, localized subsidence associated with geological, morphological and/or anthropogenic features, collapse instabilities related to cavities excavated in soft rocks, for the extraction of construction material, and sinkholes. When anthropogenic cavities are located in urban areas, their stability assessment could represent a challenging problem because they potentially induce deformations of the overlying ground surface, affecting the structures and infrastructures. Therefore, cavities represent one of the major hazards to building heritage. From an analytical point of view, the extent of subsidence influence at the surface level, related to the presence of a cavity, is determined by considering the angle of draw [24]. Sinkholes are the manifestation of collapse at the ground surface, where water drains naturally. The collapse of sinkholes could have both natural and anthropic causes and these are difficult to predict. The occurrence of sinkholes can be frequent where soft rocks lie close to the ground surface and weathering process creates voids that determine the conditions favourable to collapse and widening [25].

In this work, we propose a new multi-method approach to analyze the main trends and seasonal signals in ground displacement time series, to understand spatial-temporal patterns, to correlate ground deformation phenomena with geological and anthropogenic factors, and to recognize specific footprints of different ground deformation phenomena. We combine PCA and ICA techniques both in temporal and spatial modes, and the obtained results are analyzed and grouped using Hierarchical Clustering (HC). This approach, which allows us to classify ground deformations at site and local scales, was applied to the metropolitan area of Naples (Southern Italy) characterized by the overlapping of both natural and anthropogenic phenomena. The data used in our analyses are taken from four InSAR datasets (TRE Altamira) covering the period from 1992 to 2019.

## 2. Materials and Methods

In the following section, the metropolitan area of Naples (Southern Italy) is described considering the geological and hydrogeological contexts in order to characterize the origin of the deformation mechanisms. Subsequently, the considered dataset of cavities is presented, and finally, the used InSAR datasets are illustrated.

### 2.1. Geological and Hydrogeological Settings

We applied our approach to the UNESCO World Heritage site of Naples (Figure 1), which is part of the historic Centre of Naples, located in the South-Central sector of the wide alluvial Campanian Plain. The Campanian Plain was formed in the Pliocene–Pleistocene by the filling of a regional half-graben with a 2000–3000 m thick sequence of Quaternary continental, alluvial, marine, fluvio-palustrine sediments and volcanic deposits [26]. In fact, starting from 300 ka, the plain has been affected by intense volcanic activity that led to the formation of the Phlegraean Fields volcanic district and Mt. Somma-Vesuvius stratovolcano. The morphology of the surrounding area is predominated by hills, related to the Phlegraean Fields, an active volcanic area located in the western sector of the city of Naples, and to the formation of the following two calderas: the Campanian Ignimbrite (CI, ~39 ka old) and the Neapolitan Yellow Tuff (NYT, ~15 ka old) [26–28].

The Phlegraean Fields experience bradyseismic phenomena [29] with considerable slow vertical ground movements, causing recurring episodes of uplift and subsidence in the order of meters or tens of meters, and accompanied by increases in shallow seismicity [30,31]. Refs. [32–35] investigated the history of the recent Phlegraean Fields unrest caldera and overviews of the ground deformation patterns. Regarding the slow-rate ground deformation processes that occurred over the period from 1992 to 2010, characterizing the Campanian region, Ref. [36] carried out their quantitative analysis and classification.

Within the municipalities of Naples, the Vomero-Arenella (VA) district, which is a pyroclastic hill, experienced subsidence phenomena attributed to its geological structure, as suggested by [37]. In particular, two main areas of subsidence affect the VA district, one bounded by the CI caldera faults, with the other located at the margin of the NYT caldera [37,38]. These faults are still active and sensitive to the dynamics of the Phlegraean Fields volcano, as suggested by an attenuation of subsidence within the VA area during the uplift of the central sector of the Phlegraean Fields [39]. Considering the hydrogeological setting, the groundwater circulation is mainly due to both fractures in tuff and pores within incoherent pyroclastic and alluvial deposits. For this reason, the hydrogeological setting of the area is both vertically and laterally complex and the water table is generally found at variable depths below the ground level.

The NYT formation, located at various depths from the ground level, dominates the geological setting of the area and it constitutes its base formation. NYT formation consists most of pyroclastic-flow and minor deposits that can occur both as lithified or not-lithified diagenetic facies. The not-lithified diagenetic facies are called pozzolana. The latter is a 10 m thick silty sand deposit, which underlies a thinner and younger formation of lapilli and pumices and preserves its primary depositional character [26]. NYT and pozzolana are characterized by fair mechanical parameters and low specific weights. The top of this pyroclastic sequence is constituted of volcanic fly ashes, remoulded soils, and artificial ground.

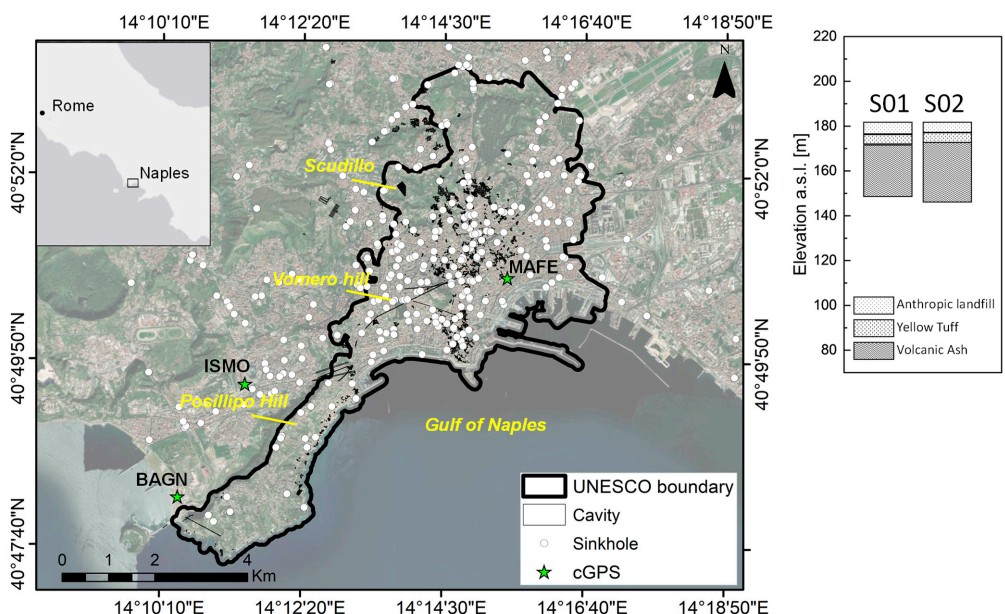

**Figure 1.** Location of the UNESCO World Heritage site of Naples (black line), which covers 31.3 km². The distribution of cavities and sinkholes [26,40–44] and the location of the considered cGPS stations [30,32,45,46] are also shown. The main considered zones of the Naples metropolitan area are indicated in yellow, as well as the location of the Scudillo water reservoir. The stratigraphy from two well logs (S01 and S02) located close to the water reservoir (adapted from [37]) is shown on the right.

### 2.2. Neapolitan Cavity System Description

The anthropogenic cavities system in the city of Naples (Figure 1) is significantly affected by sudden collapses that generate damage to infrastructure, buildings, and cultural heritage [40]. This system has been mainly excavated through the NYT and loose pyroclastic soils in the last 2000 years and it is mainly composed of tunnels and chambers of various sizes and geometries depending on their different purposes. Those cavities have been frequently used as dumping sites for waste materials or, in other cases, as reservoirs for sewage systems. It is worth noting that a large number of them are still unknown.

Sinkholes are mainly located in the central part of the area and close to the historical centre where steep slopes' morphology, ancient buildings and high structures' density occur. Refs. [26,40–44] collected and analyzed data on sinkholes in this area and identified rainfall, sewerage, and drainage system leaks as the main triggering factors, specifying that their influences are not uniform within the different districts of the Naples area. One of the most important cavities is the Scudillo water reservoir (SC,) whose area is characterized by subsidence and a negative deformation rate increasing over time, with maximum values of approximately −8.23 mm/year, recorded by the TerraSAR-X dataset in the 2015 ÷ 2019 period (TRE Altamira). The SC area is a sparsely built-up zone characterized by a morphological bulge delimited by lateral incisions. The local stratigraphy [37] shows 4 m of reworked pyroclastic deposits and pumices levels covering cohesive NYT deposits between 6 and 8 m thick (Figure 1 on the right). The SC is a large artificial network of cavities, built within the NYT, approximately 40 m below the ground level and measuring about $350 \times 150 \times 10$ m with a surface area of $5.25 \times 10^4$ m². These cavities are currently used as a drinking water reservoir, with an estimated capacity of 145,000 m³. According to [37], the tunnels surrounding the main reservoir show that the NYT is affected by fractures. They exclude that the subsidence of the SC is triggered by hydrological causes since the groundwater table is located below the cavity (between 14 and 18 m a.s.l.), as well as by faulting activity or by soil softening. They suggest that subsidence could be related to the artificial cavity and to the general gravitational instability induced by the morphological bulge.

*2.3. InSAR Datasets*

The purpose of this section consists of describing the characteristics of the datasets used in the following analysis to highlight their technical features.

InSAR is a highly effective remote sensing technique used to monitor changes in the Earth's surface. This technique involves the acquisition of radar images from a satellite or an airborne system over the same area, but at different time frames. By measuring the phase shift of the signal between two images, InSAR can detect even the slightest movements of elements located on the Earth's surface. This displacement is measured along the direction of the radar sensor-target line, also known as the Line-Of-Sight (LOS) of the satellite. The result of this process is a series of images that effectively represent the monitored portion of the Earth's surface, enabling researchers and scientists to analyze and interpret changes in topography, infrastructure, and natural phenomena with unprecedented precision and accuracy.

To quantitatively determine the value of ground displacement from the acquired InSAR images, one of the most widely used techniques reported in the literature is Permanent Scatterer SAR Interferometry (PSInSAR) proposed by [1], which was the first of a number of algorithms commonly referred to as PSI techniques. All these processing algorithms use multi-image datasets to identify stable reflectors called Permanent or Persistent Scatterers (PS). PS are radar targets located on the Earth's surface that reflect stable signals back to the satellite sensor and exhibit stable amplitude and coherent phases over long temporal series of images. They are typically elements of buildings, metallic structures and infrastructures, as well as boulders and rocky outcrops.

While PSI algorithms aim at identifying pointwise coherent targets, other techniques have been developed to increase the density of measuring points over non-urban areas, exploiting the so-called distributed scatterers (DS), which are clusters of image pixels exhibiting very similar radar returns (e.g., Small BAseline Subset, SBAS, [47]). More recently, new algorithms have been developed by extracting information from both PS and DS. SqueeSAR® proposed by [48] enables us both to increase target density in non-urban areas, and to more effectively filter out atmospheric disturbances and noise. The most influential factors affecting the quality of the measurements, and therefore of the results obtained, are the spatial density of the measurement points, the quality of radar targets, the atmospheric conditions at the acquisition time, the distance between the measurement points and the reference points (REF) and the number and temporal distribution of acquisition. Regarding the precision obtained by the PSInSAR™ technique, in terms of differential displacement measurements, the reported value is ±5 m and considering the mean velocity, the value is ±1 mm/year [1]. For measurements obtained using the SqueeSAR™ technique [48], precision is defined by considering a dataset of at least 30 scenes covering a two-year period, for a measurement point located less than 1 km from the REF. Precision is expressed in terms of standard deviation, which refers to the average displacement rate relative to the REF. The typical accuracy obtained from SqueeSAR™ analysis is lower than 1 mm/year for average annual velocity, while the single measurement is generally within ±5 mm/year [49].

To investigate the spatiotemporal evolution over 27 years of ground deformations in the metropolitan area of Naples, four datasets acquired from different SAR satellites (ERS1-2, ENVISAT, COSMO-SkyMed, and TerraSAR-X) were analyzed. InSAR measurements were obtained from C-band and X-band radar sensors in descending geometry and were processed using PSInSAR and SqueeSAR® techniques by TRE Altamira. Data acquired by ERS1-2, ENVISAT and COSMO-SkyMed satellites were processed using a PSInSAR approach, whereas TerraSAR-X data were the result of a SqueeSAR® analysis. The ERS1-2, ENVISAT and COSMO-SkyMed datasets cover the UNESCO World Heritage site of Naples (31.3 km$^2$) whereas the TerraSAR-X dataset extends over a 325.8 km$^2$ area (Figure S1). Table 1 summarizes the main features of the InSAR datasets. The InSAR data analyzed, with regard to the COSMO-SkyMed and TerraSAR-X datasets, are the results of SAR image datasets formed by stacks of StripMap (SM) images. The SM is the basic SAR imaging mode. The ground swath is illuminated with a continuous sequence of pulses while the antenna

beam is fixed in elevation and azimuth. This produces an image strip with continuous quality (in the direction of flight). SM data with double polarization, such as those from which the analyzed data are derived, have a slightly lower spatial resolution and a smaller swath than the single polarization data. In SM mode, a spatial resolution of up to 3 m can be achieved. Twin polarization SM data recorded in HH (single polarization channel) have a standard scene size of 30 × 50 km (width × length). Specifically, in this work, the area of the UNESCO World Heritage site of Naples was analyzed, which spans a surface of 31.3 km$^2$ in total (Figure 1).

**Table 1.** Main features of the InSAR datasets used for the analyses.

| Satellite | Sensor's Band | Orbit | Inc. Look Angle (°) θ | Acquisition Span | Area (Km$^2$) | No. PS-DSs | Mean PS-DSs Density * | Spatial Resolution (m) * | Revisit Time (Days) |
|---|---|---|---|---|---|---|---|---|---|
| ERS1-2 | C | Desc. | 23 | Jun 1992 Dec 2000 | 31.3 | 8122 | 262 | 20 × 5 | 35 |
| ENVISAT | C | Desc. | 23 | Jun 2003 Jun 2010 | 31.3 | 15,380 | 496 | 20 × 5 | 35 |
| COSMO-SkyMed | X | Desc. | 44 | Feb 2012 Dec 2013 | 31.3 | 252,977 | 8160 | 3 × 3 | 8 |
| TerraSAR-X | X | Desc. | 21.6 | Jan 2016 Apr 2019 | 325.8 | 2,566,269 | 7876.8 | 3 × 3 | 11 |

* The PS-DSs density stands for the number of PS-DSs divided by the area in km$^2$, whereas the spatial resolution is the azimuthal range.

As previously mentioned, InSAR displacements are measured along the satellite LOS. Since the aim of this study is to describe and classify vertical displacement mechanisms, the measured velocities and displacements of each InSAR dataset were projected along the vertical direction. In this work, we only considered the vertical component of the displacement as a consequence of the available dataset. Since only the descending dataset was available, it was not possible to calculate the east–west component of the displacement. However, we mainly focused on uplift, subsidence and sinkholes phenomena, for which the main component of the displacement is the vertical one. To achieve this, the velocities and displacements along the LOS were divided by the cosine of the local incidence angle θ (i.e., the angle between the LOS and the vertical direction—see values in Table 1). In particular, local scale analyses focused on small cavities and sinkholes, where the horizontal component of displacement is not the main component, even if useful for further investigations. An exception to this is the Phlegraean Fields area, where bradyseism phenomena cause both large vertical and horizontal displacements, and both components must necessarily be considered. A possible shortcoming of not considering the horizontal component consists of neglecting other processes (e.g., landsliding) that are not considered in this work. However, the proposed methodology can be applied both to vertical and east-west displacement time series.

All displacement time series, projected along the vertical direction, were then linearly interpolated to resample them with regular acquisition in time. Linear interpolation is used to handle missing values that may occur in the time series and consequently to fill the data gaps beforehand. However, as a few dates are generally missing, this interpolation does not alter the results of the proposed analysis. Apart from the possible artifacts and noise, the obtained time series are affected by uncertainties related to possible atmospheric leakage, regional trends, and possible anomalous displacements occurring on specific dates. To overcome these artefacts, we employed the detrending approach proposed by [50], which consists of (i) selecting the most coherent measurement points (coherence greater than 0.9) with an average LOS velocity in the range ±0.05 mm/year; (ii) computing the average time series of the selected PS-DS and (iii) subtracting it from all the projected time series.

Figure 2 illustrates the mean vertical deformation rate maps obtained from the time series after the detrending approach. In agreement with the mean vertical deformation rate

maps, Figure 2a illustrates the following three main subsidence zones detected by ERS1-2, between 1992 and 2000: Vomero-Arenella (VA), Scudillo (SC), and Posillipo (PO) districts. Figure 2b,c display the mean vertical deformation rate maps over the period monitored by Envisat and COSMO-SkyMed, from 2003 to 2010 and from 2012 to 2013, respectively. The maximum mean vertical deformation rates are observed in the southern area, within the PO district, from 2003 to 2013, indicating that the area was affected by uplift, and thus modifying the subsidence pattern resulting from the ERS1-2 dataset. Considering only the UNESCO site, Figure 2d illustrates the mean vertical deformation rate related to the TerraSAR-X dataset between 2016 and 2019. Figure 2d shows that the main subsidence zones during the monitored period were located in the VA and SC districts as confirmed by previous studies [37]. Finally, Figure S1 (Supplementary Materials) illustrates the mean vertical deformation rate maps of 10,000 PS-DSs randomly selected from the TerraSAR-X dataset with the greatest uplift in the south-western area, within the Phlegraean Fields, exceeding 5 mm/year. The highest subsidence occurred in the central area of the UNESCO site, within the VA district, with a rate exceeding −5 mm/year. According to the mean vertical deformation rate map, it is observed that the PO area, after the period monitored by ERS1-2 (1992–2000), during which the area was affected by subsidence, shows a continuous upward trend, gradually expanding northward. Figure 3 shows two times series of LOS displacement from the TerraSAR-X dataset at the SC and PO zones.

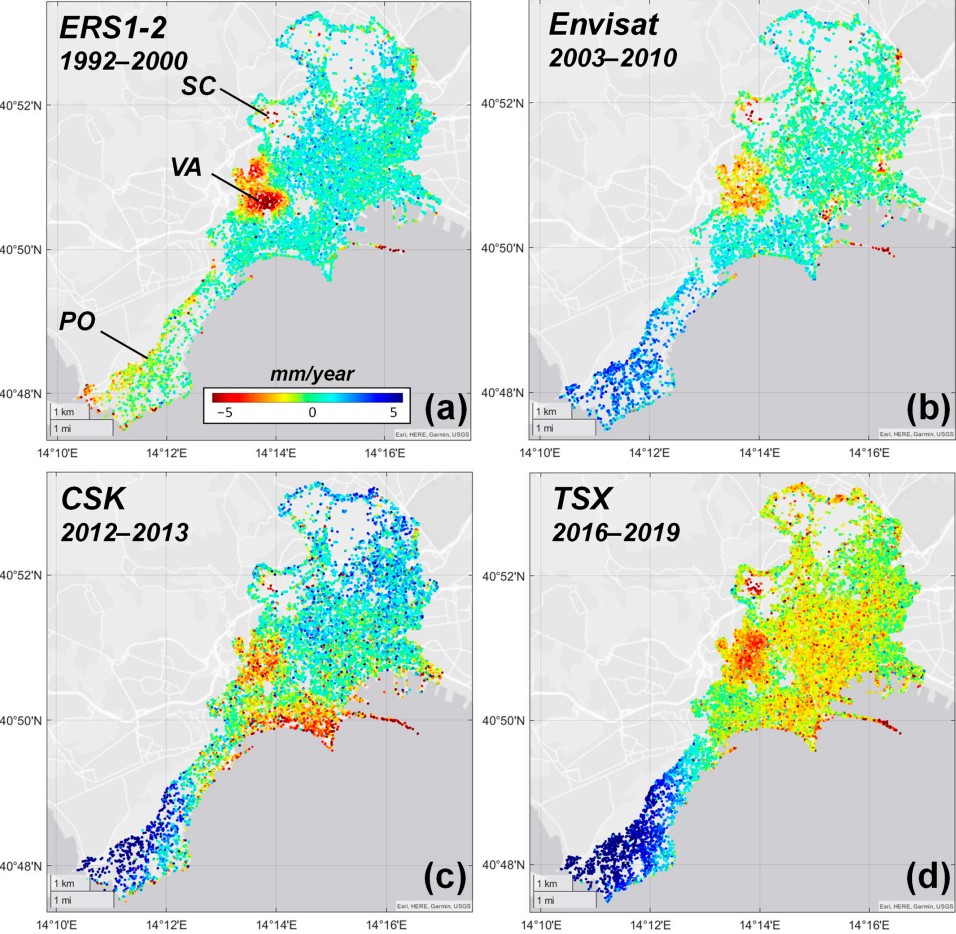

**Figure 2.** Maps of the mean deformation rate in the vertical direction obtained from the pre-processing procedure for (**a**) ERS1-2 spanning from 1992 to 2000, (**b**) for Envisat spanning from 2003 to 2010, (**c**) for 10,000 PS randomly selected from COSMO-SkyMed spanning from 2012 to 2013, (**d**) and for 20,000 PS-DSs randomly selected from the TerraSAR-X dataset covering the UNESCO site spanning from 2016 to 2019.

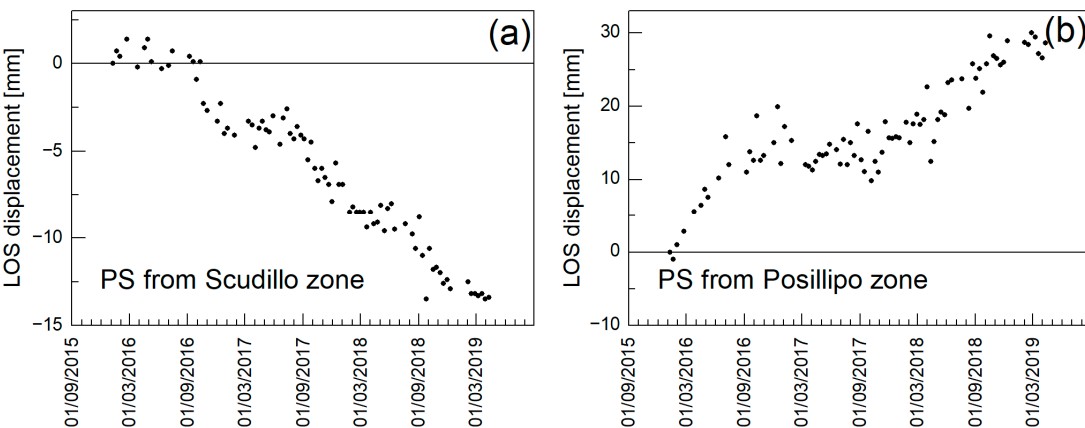

**Figure 3.** Temporal evolution of the LOS displacement obtained from TerraSAR–X for (**a**) one PS-DS at the Scudillo zone and (**b**) one PS-DS at the Posillipo zone.

To validate the InSAR data used in this work, the satellite-based ground measurements were compared with the displacements recorded by cGPS stations related to the Neapolitan Volcanoes cGPS (NeVoCGPS) network, operated by the Istituto Nazionale di Geofisica e Vulcanologia-Osservatorio Vesuviano (INGV–OV). For this purpose, we used cGPS measurements acquired on a weekly basis between January 2012 and March 2020. We considered the cGPS MAFE, ISMO and BAGN stations operating in the Phlegraean Fields area [18,45,46], as shown in Figure 1. To validate the InSAR data, we considered the vertical component time series of the three cGPS stations and compared them with the COSMO-SkyMed and TerraSAR-X time series, projected in the vertical direction as explained in this section. The comparisons between InSAR data and cGPS show a reasonable match between the two datasets (3 mm error bands for the cGPS measurements are given in Figure S2).

*2.4. Methodology*

The workflow of the methodology employed in this study is shown in Figure 4 and involved the following four main steps: (i) time series pre-processing; (ii) S-mode and T-mode PCA; (iii) S-mode and T-mode ICA; (iv) clustering analysis applied to the PCA and ICA scores. A self-built code in MATLAB (R2021a, The MathWorks, Inc., Natick, MA, USA) was implemented to perform all the analyses.

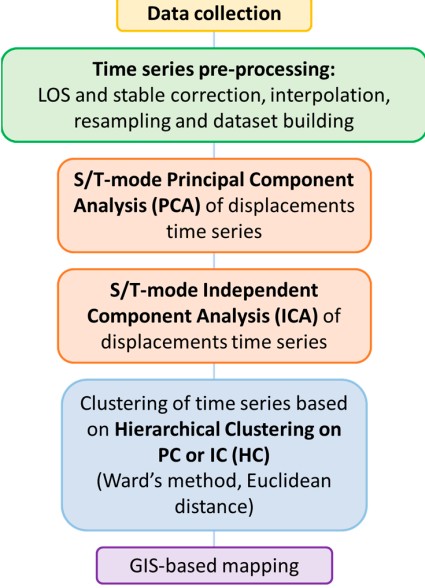

**Figure 4.** Methodological flowchart representing the proposed algorithm.

### 2.4.1. PCA Decomposition

The time series of ground displacement are the results of the superimposition of many signals generated by different phenomena, acting simultaneously at different scales. Several approaches have been proposed in the literature to separate and analyze these signals. The purpose of this section is to illustrate the methodology used to achieve the above-mentioned goals. The Principal Component Analysis (PCA, [51–53]) is probably the most widely used multivariate statistical technique [2,54,55]. This methodology assumes that the recorded time series is a linear combination of the original signals associated with different phenomena and attempts to reduce a dataset containing a large number of variables to a dataset containing fewer new variables [56]. The separation is performed by projecting the original signals into a reduced hyperspace spanned by orthogonal vectors that are the eigenvectors of the covariance matrix of the original null-mean signals. The corresponding projections are the principal components only if the corresponding eigenvalues of the covariance matrix of the null–mean signals are positive. This is because the eigenvalue of a principal component is its variance and for each principal component, the explained variance, ExVar, is computed using the following expression:

$$\text{ExVar}(y_i) = \frac{\lambda_i}{\sum_{j=1}^{n_c} \lambda_j} \qquad (1)$$

where $\lambda_i$ is the eigenvalue of the *i*-th principal component $y_i$ and $n_c$ is the number of principal components extracted from the analysis.

In this way, PCA transforms a collection of original intercorrelated variables to a new limited set of uncorrelated variables sorted according to their explained variance. Not all the principal components are considered, as the first components often explain most of the variance. Therefore, PCA is a data reduction method, in which most of the information of a dataset can be reproduced by a reduced number of components [11]. The correlation between any one of the original variables and a component is called loading (eigenvectors) and the values associated with each sample are called principal component scores, which are the new coordinates of the observations in the principal component hyperspace.

PCA can be applied to the dataset in the following two ways: the T-mode and S-mode. In the T-mode, which is the most commonly used [2,3,57,58], the PCA is applied to a data matrix X defined by n by p, where the n rows of X correspond to each observation and the p columns to variables. In this study, the variables correspond to acquisition times and observations correspond to each PS-DS; therefore, the resulting covariance matrix is p by p. In the S-mode, the PCA is applied to the transposed n by p data matrix X, so that the p rows of X correspond to variables and the n columns to observations. In the S-mode, the variables correspond to acquisition times and observations correspond to each PS-DS. In other words, in the T-mode, the variables are temporal instants, whereas in the S-mode, the variables are temporal profiles (time series). Due to the different configurations of the data matrix, T-mode PCA finds consistent patterns in space, whereas S-mode PCA detects spatially significant patterns. This is related to the orientation of the input time series matrix, as by centering across the columns of the matrix, and thus averaging the individual time series, each PS-DS has the same weight no matter its geographical location. This capability is due to the fact that, unlike the S-mode, in the T-mode matrix configuration, the centering is performed on a date-by-date basis, averaging the individual time instants. By default, the PCA function implemented in MATLAB centers the data and uses the singular value decomposition (SVD) algorithm.

### 2.4.2. ICA Decomposition

The second technique applied in this study is the Independent Component Analysis (ICA), based on the mathematical assumption that the original signals are correlated to the source signals, and that this mixture is described by means of the following equation:

$$R(t) = MS(t) + N(t) \qquad (2)$$

where $R(t)$ is the matrix that collects the original signals (time series) arranged by rows. In the above equation, $M$ is the mixing matrix representing the superimposition method of the source signals and indicating how the individual source signals are combined to obtain the original signals, and $S(t)$ is the matrix representing the source signals of each phenomenon occurring during the data recording. The last term in the equation, $N(t)$, is the matrix expressing the noise embedded during the recording phase, not resulting from a specific phenomenon, and demonstrating Gaussian distribution. In the ICA, noise is assumed to be a statistical variable that follows a Gaussian pattern and the method to separate the original signals is based on maximizing the non-Gaussianity of the signal sources. Behind this technique, there is the principle that if the signals of the sources are physically different, then they are statistically independent. For this reason, the ICA technique attempts to decompose the set of original signals into a set of statistically independent components. Therefore, the probability density functions of the recorded data are the product of the probability density functions of the source signals because they are non-Gaussian. The decomposition is performed by extracting source signals that have a probability distribution as distant from the Gaussian distribution as possible. The algorithm underlying ICA finds source signals that maximize non-Gaussianity using the following two possible ways: maximizing negentropy or kurtosis. In contrast to PCA that maximizes the signal according to the variance, which requires the high importance of the signal, this approach is able to detect and also extract low-intensity signals because these are statistically independent. In this work, the fast fixed-point algorithm for ICA (FastICA, [59]) was applied. The FastICA algorithm involves the following five steps: (i) centering the data by subtracting the mean from the mixed-signal matrix to obtain zero mean variables; (ii) calculating the eigenvalues and eigenvectors using the PCA algorithm; (iii) calculating the whitening matrix; (iv) whitening the data; (v) computing the directions of the independent components from the whitening.

### 2.4.3. Hierarchical Clustering

Hierarchical Clustering (HC) analysis [60] consists of a progressive clustering of a set of n elements starting from a number of sets equal to n, containing only one element, and a progressive union of them based on the minimization of a defined objective function. This approach has been used in the literature to analyze meteorological data [12,61] and can be applied to any variable and any objective function. In this work, we applied HC to the scores of the principal and independent components obtained from the PCA and ICA and used the total inertia of all clusters as the objective function. To prevent components with a higher order of magnitude from hiding components with low magnitude, the score components considered in the cluster analyses were initially standardized by the scaling method, i.e., subtracting the mean value from the scores and dividing them by their standard deviation. This is important because dissimilarity between observations is calculated as the statistical distance between them and since distances are unit-sensitive, cluster solutions may change when the data are scaled. It is important to note that the clustering is based on the values of the scores and not on the spatial distribution of the coordinates of the PS-DSs, although the representation of the clusters is then spatially visualized. To evaluate the optimal number of data clusters to be created, the Silhouette approach was applied and the dendrogram was considered.

## 3. Results

Site scale analyses were performed at the UNESCO World Heritage site scale ($31.3 \text{ km}^2$), to identify the main temporal and spatial patterns of ground deformation, whereas local scale analyses were performed at the cavity scale to classify the ground deformation related to the presence of cavities.

For the site scale analyses, the InSAR time series of each dataset were analyzed by PCA and ICA multivariate statistical analyses in both spatial (S-) and temporal (T-) modes. The entire ERS1-2 and Envisat time series datasets were analyzed, while the COSMO–SkyMed

and TerraSAR-X datasets were spatially randomly sub-sampled into 10,000 and 20,000 time series (PS-DSs), to reduce the computational burden. Down sampling was conducted by selecting uniformly distributed random PS-DSs from the original dataset using the rand function in MATLAB.

For the local scale analyses, we performed T-ICA and clustering analysis on the TerraSAR-X dataset, considering the PS-DSs belonging to all buffers surrounding each cavity.

### 3.1. S- and T-PCA and ICA Results for TerraSAR-X Dataset at Site Scale

From PCA, in both the S-mode and T-mode, we extracted ten components, based on the eigenvector explained variance, from which it was decided how many components of these to retain for S-mode and T-mode ICA. This was performed based on the eigenvalues (variance), the trend of the temporal functions (eigenvectors, to avoid noise), and the spatial distribution of the retained components (scores, useful for clustering). In this section, the results obtained only from the TerraSAR-X dataset are reported. The results refer to the 20,000 time series (PS-DSs) covering only the UNESCO site. We initially present the S-mode results, as they can identify the components that cover a large number of PS-DS of the study area. S-mode PCA was applied to the dataset and the first four components were considered accounting for 49.28%, 3.95%, 3.43%, and 2.23% of the total variance (Figure 5a). Based on the inspection of the temporal functions of the extracted components (Figure 5b), PC1 is characterized by a decreasing trend and PC2 and PC3 by seasonal behavior, PC3 has peaks in the summer and PC2 is in the opposite phase to PC3, and PC4 does not show a clear temporal trend.

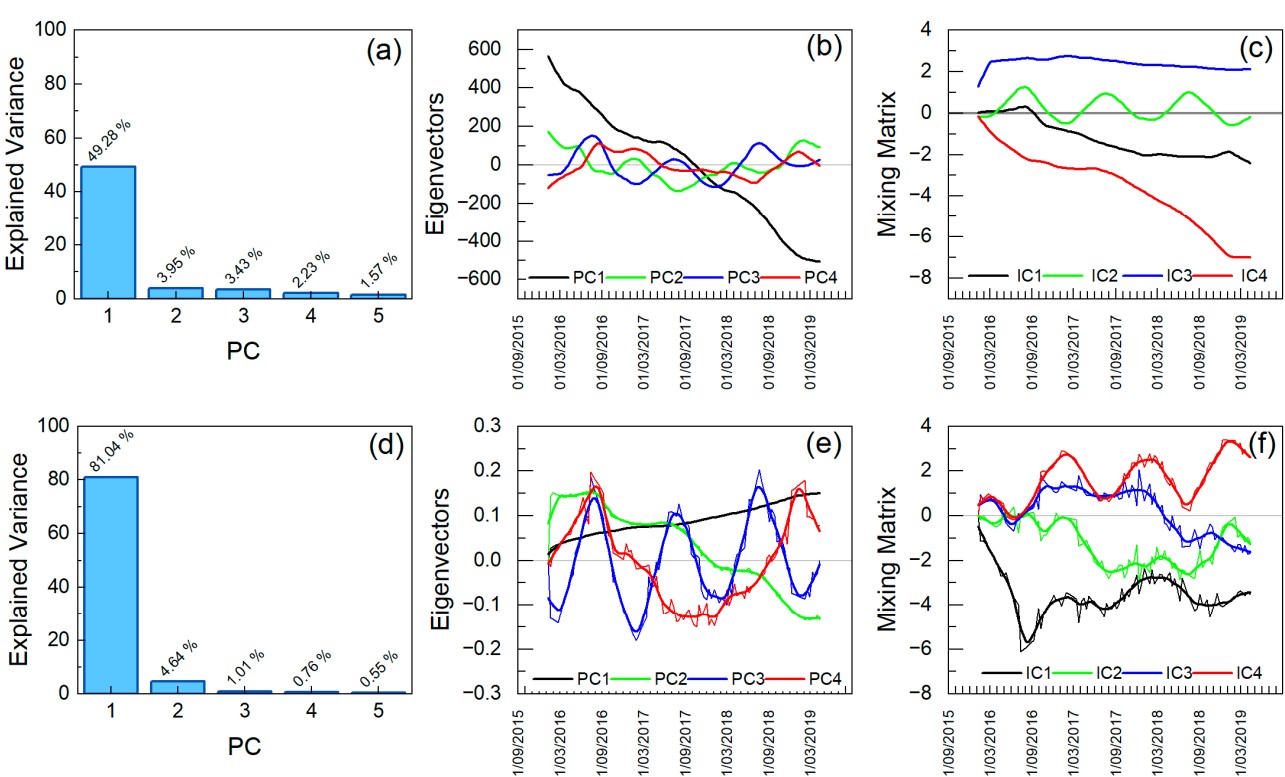

**Figure 5.** (**a**,**d**) Explained variances for each PC from the S- and T-mode, respectively. Temporal functions of components extracted from (**b**) S-PCA (**c**) S-ICA, (**e**) T-PCA, and (**f**) T-ICA, on 20,000 randomly selected PS–DSs from TerraSAR-X within the UNESCO site. The smoothed lines were obtained from the results using the lowess method with 0.1 span.

Following the proposed algorithm (Figure 4), S-mode ICA was performed by retaining the first four PCs, and among the four obtained ICs, we considered IC2 and IC4 as they present a clear temporal trend. Based on their temporal functions (Figure 5b,c), IC4 de-

scribes a decreasing trend, while IC2 demonstrates regular seasonal behavior. Considering the spatial distribution of the extracted components, PC1 and IC4 correspond to the long-term subsidence and are correlated with the mean vertical deformation rate with positive scores (Figure S3).

These components highlight the VA and SC water reservoir areas that are affected by subsidence during the period monitored by TerraSAR-X, and with negative values highlight the Southern area of PO, which is affected by uplift. PC2, PC3 and IC2 components represent the seasonal signals characterizing the VA and SC districts affected by annual deformations in phase with PC3 and IC2 and at the same time highlighting the coastal zone in the central part of the UNESCO site where seasonal deformation in the opposite phase to IC2 signal is observed.

After investigating the trends that characterize most of the study area, T-mode PCA was applied, as the first components to be retrieved are those that best describe the displacement rates. Based on the T-mode PCA results, we considered the first four PCs accounting for 81%, 4.63%, 1%, and 0.76% of the total variance (Figure 5d). The eigenvector temporal functions (Figure 5e) of the components define in PC1 an increasing trend, in PC2 a decreasing trend and seasonal behavior (peaks in summer), in PC3, seasonal behavior in phase with PC2, whereas PC4 has a non-regular trend, with two peaks, in August 2016 and January 2019. As for the S-mode analyses, T-mode ICA was applied to the dataset considering the first four PCs and based on the temporal functions of the mixing matrix (Figure 5f). IC1, IC2 and IC3 define decreasing trends, while IC4 demonstrates increasing seasonal behavior with peaks in winters. Based on the scores map (Figures S4 and S5), PC1 and PC2, as well as IC2 and IC3, display the same pattern of scores distributions, which is correlated with the mean vertical displacement, defining the following two main zones of subsidence emphasized by positive values of ICs: the SC water reservoir and VA district; and with negative values, the area of PO. IC1 is strictly related to the ground deformation occurring in the southern area of PO with negative values as for PC4. In fact, the fourth PC with positive scores is only related to the deformation occurring in the most Southern area of PO. According to the seasonal components, PC3 and IC4 are related to the seasonal ground deformation and to the decreasing trend occurring in the VA and SC, where the deformations are in phase with the extracted PC3. PC3 and IC4 are also related to the central coastline sector where the deformations are in opposite phases to the extracted signal, and to the PO district where signals are in phase with IC4.

## 3.2. S-ICA Results for All Datasets at the Site Scale

Since S-ICA was found to be the most efficient technique to identify and extract ground deformation patterns at the site scale, in this section, the results obtained by applying this technique to all InSAR datasets are detailed. In general, three to six components were extracted from the S-ICA. However, based on the results, two to four components were considered among those extracted. Since the results regarding the S-ICA applied to the TerraSAR-X dataset were already reported in the previous section (Figure 5c), we now consider the remaining three datasets.

We initially illustrate ERS1-2 in which we considered IC1 because it reflects an increasing linear trend and IC5, as it reflects an annual behavior (peaks in summers, Figure 6a), while the other components do not show any significant temporal trend. IC1 is the most significant trend and explains with negative scores the subsidence affecting the districts VA, and SC, and with lower rates, the southern part of the PO district. IC1 reflects the mean vertical displacement map and IC5 highlights the southern area of PO, which is affected by subsidence and shows annual deformations in phase with the component IC5 (Figure 6b,c).

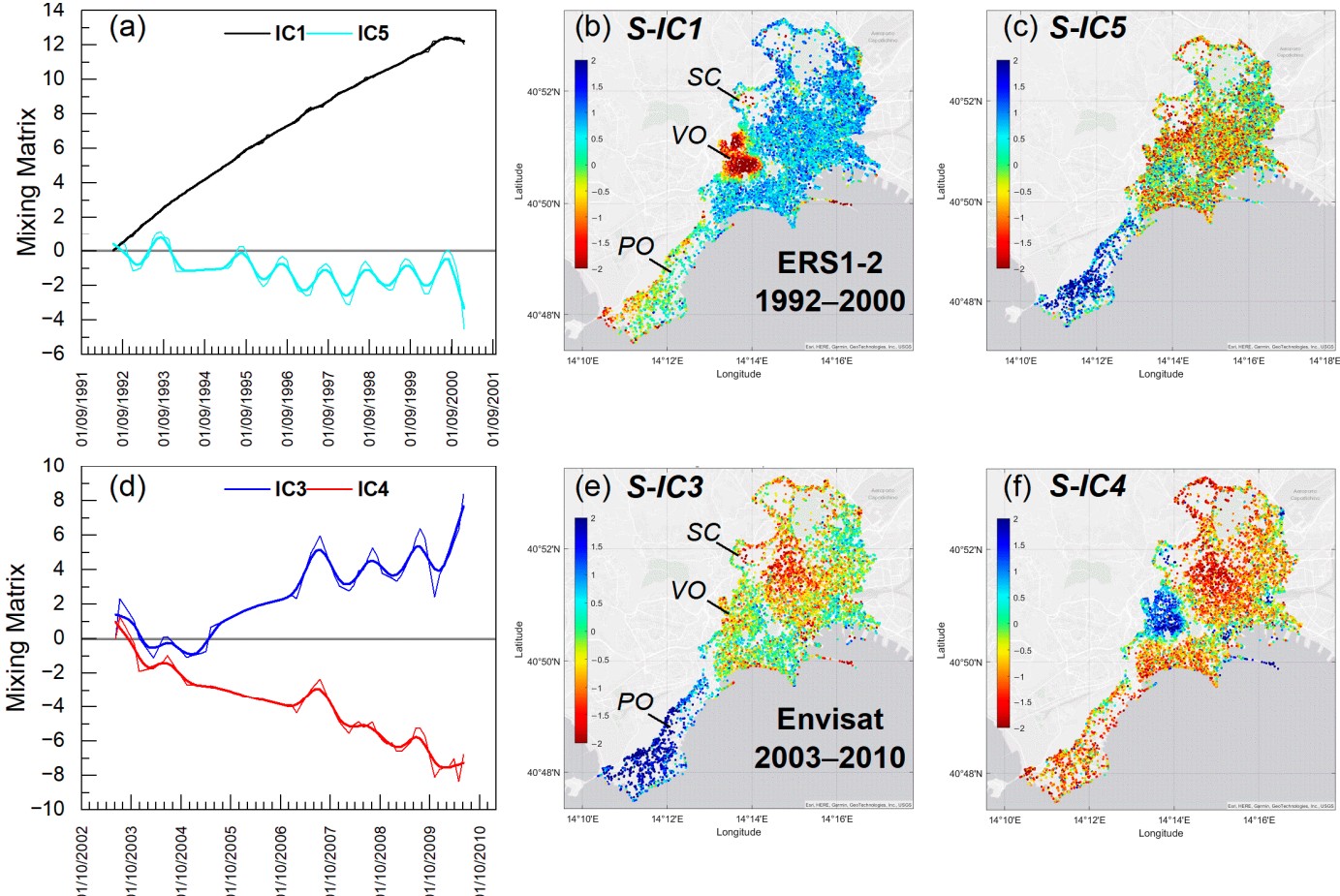

**Figure 6.** S-ICA results: (**a**) temporal evolution of S-IC1 and S-IC5 for the ERS dataset; (**b**,**c**) spatial distributions of S-IC1 and S-IC5, respectively; (**d**) temporal evolution of S-IC3 and S-IC4 for the Envisat dataset; (**e**,**f**) spatial distributions of S-IC3 and S-IC4 from Envisat, respectively. The smoothed lines are obtained from the results using the lowess method with 0.1 span.

For the Envisat dataset, components IC3 and IC4 were considered (Figure 6d), with the former demonstrating an increasing trend and seasonal behaviour (peaks in summers), and the latter demonstrating a decreasing trend and seasonal behaviour in phase with IC3. IC3 is related to the southern area of PO with positive component values; thus, it highlights uplift and seasonal movements in phase with the extracted mixing matrix time series. IC4 reflects the long-term subsidence, which emphasizes the VA district (Figure 6e,f).

Finally, for COSMO-SkyMed, three among the extracted four components were considered. IC1 shows a seasonal trend with peaks in summers, IC3 describes an increasing trend and IC4 is defined by a decreasing trend since August 2012 (Figure 7a). The PO district is highlighted with positive values of IC1 and negative values of IC4, while the VA is related to IC3 by negative values and the SC is related to IC3 by positive values. In addition, the coastal area is identified by negative values of IC3 and positive values of IC4 (Figure 7b–d).

The displacement temporal evolution associated with each component is the result of the multiplication of the temporal function of the mixing-matrix with the corresponding scores of that PS-DS. Adding these displacement temporal evolutions, only in cases of the monotonic trend (i.e., we exclude the seasonal variations), we obtained the characteristic displacement trend of the considered PS-DS. By assessing the percentage of the maximum value of these characteristic displacements in comparison with the maximum displacement measured by the satellite, we obtained different values. In cases of SC, these percentages are as follows: 51% for ERS1-2; 54% in the case of Envisat; 60% in COSMO-SkyMed, and finally, 94% when TerraSAR-X is considered.

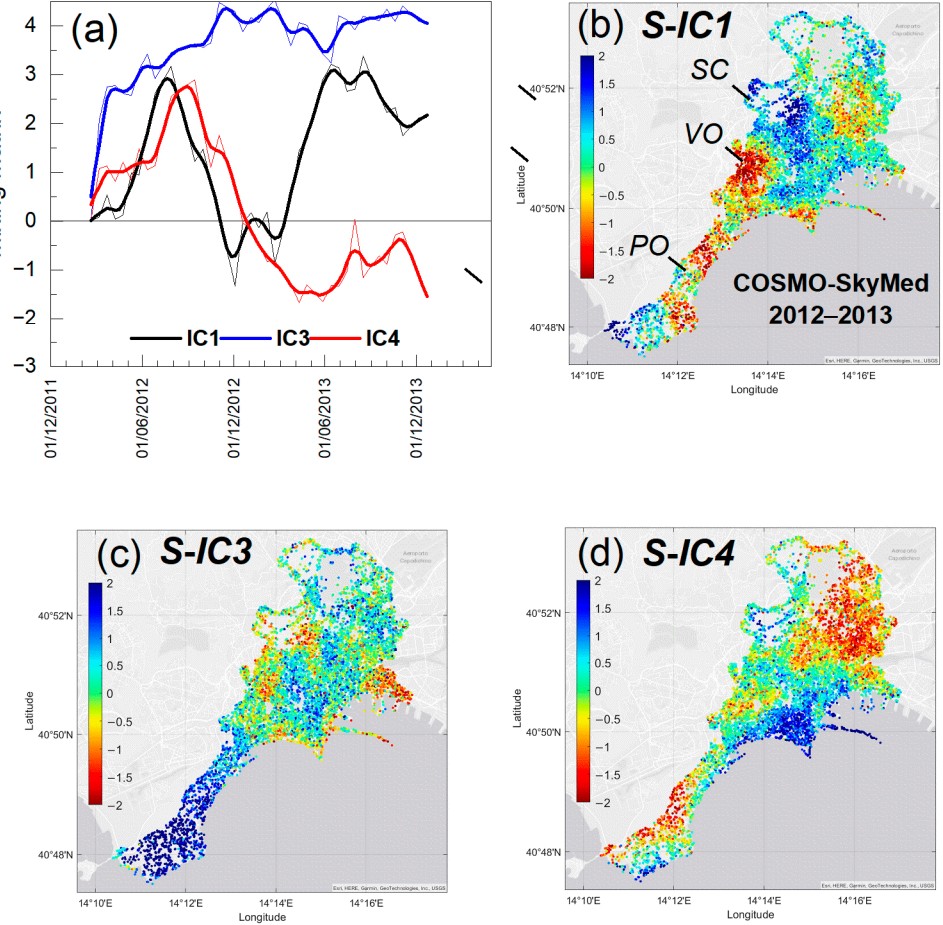

**Figure 7.** S-ICA results: (**a**) temporal evolutions of S-IC1, S-IC3 and S-IC4 from COSMO-SkyMed; (**b**–**d**) spatial distributions of S-IC1, S-IC3 and S-IC4 from COSMO-SkyMed respectively. The smoothed lines are obtained from the results using the lowess method with 0.1 span.

### 3.3. Clustering Results for S-ICA on TerraSAR-X Dataset at the Site Scale

To identify homogeneous areas of site-scale ground deformation and to further interpret the results obtained from PCA and ICA, we performed HC on the scores derived from S-PCA and S-ICA. In this section, the results of the HC analyses applied to the standardized scores of the TerraSAR-X obtained from the S-ICA are reported. HC was performed using the scores obtained from the S-mode ICA by taking into account IC1, IC2 and IC4 (Figure 8b). The results are plotted in the projected planes IC1-IC2 (Figure 8a) and IC1-IC4 (Figure 8c), obtaining four cluster IDs based on the Silhouette evaluation and on the dendrogram inspection (Figure 9). In particular, in Figure 9a, the evaluation of the optimal number of cluster data using the Silhouette method is reported, whereas in Figure 9b, the dendrogram is shown. In Figure 8a, IC1, which describes a decreasing trend, makes it possible to isolate cluster ID 2, characterized by mainly positive values, from other cluster IDs, which are characterized by negative values. Therefore, the PS-DSs that belong to cluster ID 2 are affected by subsidence, while the PS-DSs within cluster ID 4 exhibit an uplift because they are isolated by negative values of IC4. IC2, describing annual behaviour (with peaks in summer), allows us to separate cluster ID 1, defined by positive values, from the other clusters, so that PS-DSs belonging to cluster ID 1 display ground deformations in phase with the extracted IC2. Finally, in Figure 8d, the spatial distribution of the clusters is shown, in which cluster ID1 is located on the VA and in other less localized areas, including the dock district, cluster ID 4 is located on the PO, while cluster ID 3 does not define well-localized areas.

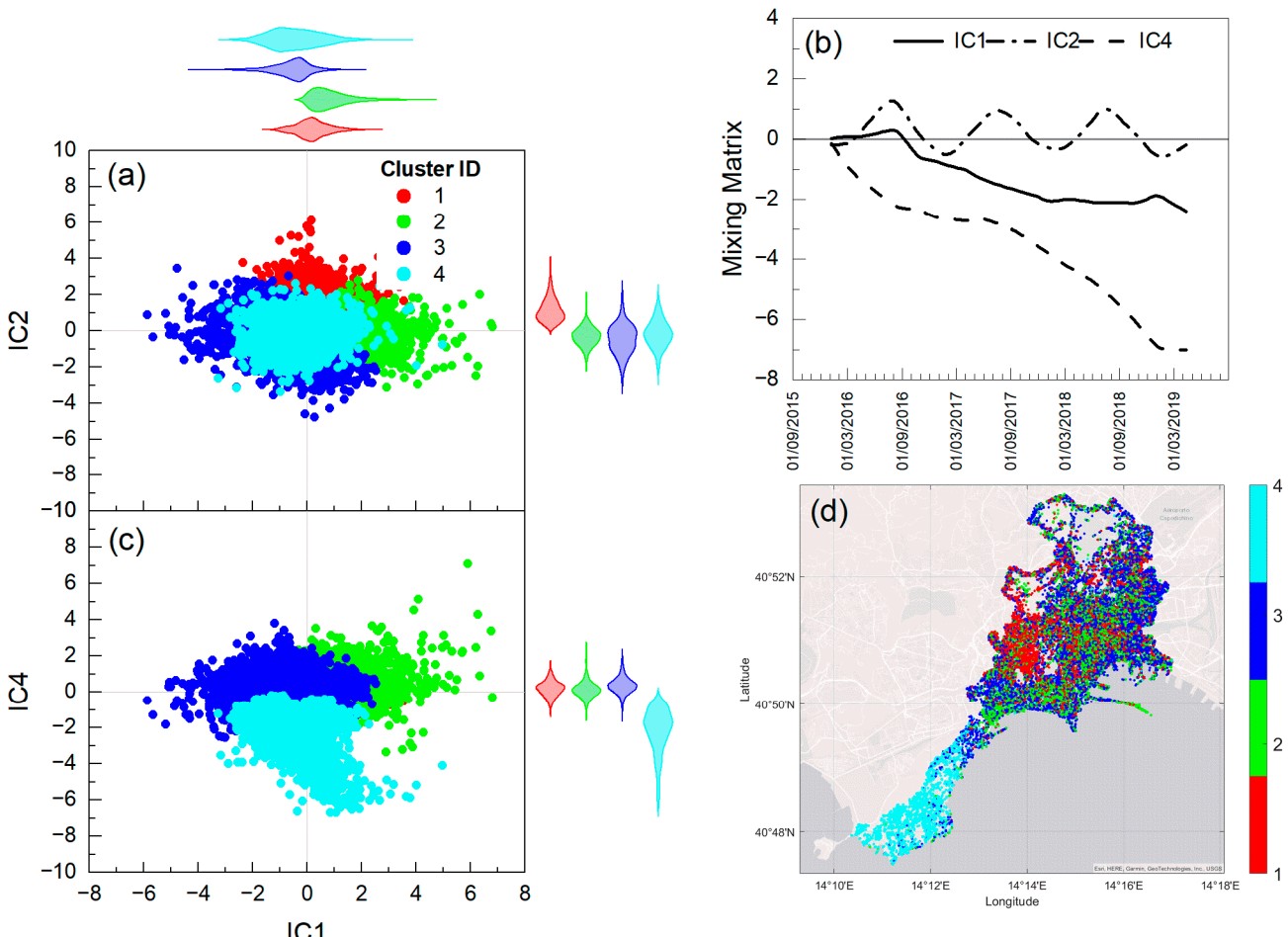

**Figure 8.** (**a**,**c**) Cluster IDs in the IC1-IC2 and IC1-IC4 planes; (**b**) S-ICA temporal functions components obtained from the spatially subsampled TerraSAR-X dataset; (**d**) cluster ID map in the UNESCO site.

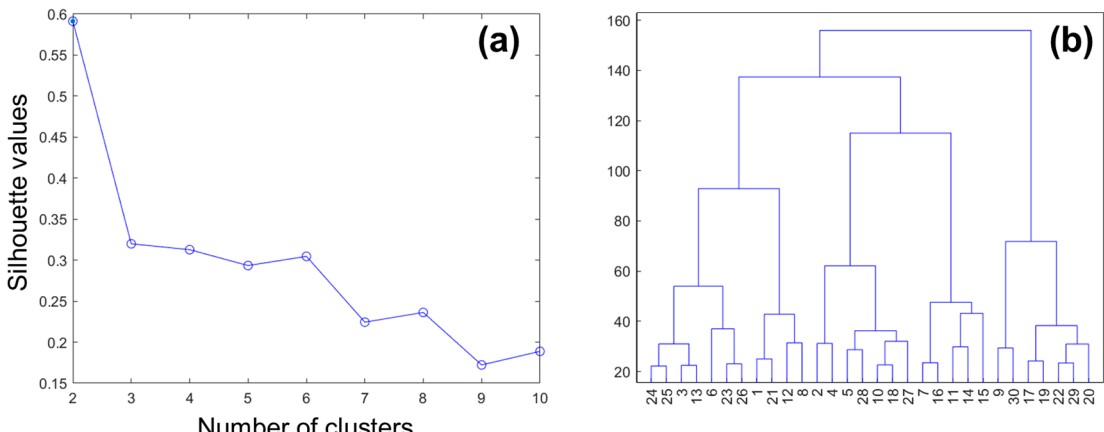

**Figure 9.** (**a**) Representation of the optimal number of clusters as a function of Silhouette values and (**b**) dendrogram obtained using 'Euclidean' distance criterion and Ward's linkage method for S-ICA spatially subsampled TerraSAR-X.

### 3.4. T-ICA and Clustering for Cavity Classification at the Local Scale

In order to classify cavities according to the ground behavior above them, T-ICA was performed considering only the PS-DS time series of the TerraSAR-X dataset that are located within all the buffers built with a 10 m offset around the cavity polygons. The

888 cavities listed in the cavity inventory [62] were considered in this classification. The buffer offset was defined based on the draw angle (i.e., the angle that defines the area of influence of the underground cavity projected onto the ground surface) [25]. We extracted four T-ICA components, as shown in Figure 10. According to the temporal functions of the independent components, we considered IC1, IC2, IC3 and IC4. Considering all the above-mentioned ICs, HC analysis was then performed following the procedure proposed in the previous sections, and five clusters were identified with different IDs, shown in the planes IC1-C2 (Figure 11a) and IC3-IC4 (Figure 11b). Moving from cluster ID 1 to cluster ID 5, PS-DSs progressively shift from being characterized by subsidence to uplift (IC3-IC4 planes). From the IC1 to the IC2 plane, we can assert that seasonality significantly influences cluster ID 5. The statistical and the 95% confidence ellipse parameters of each cluster are reported in Table S1. The centroid of each cluster, corresponding to a PS-DS, was identified and the original time series was compared with that reconstructed from the considered four ICs, as shown in Figure S6.

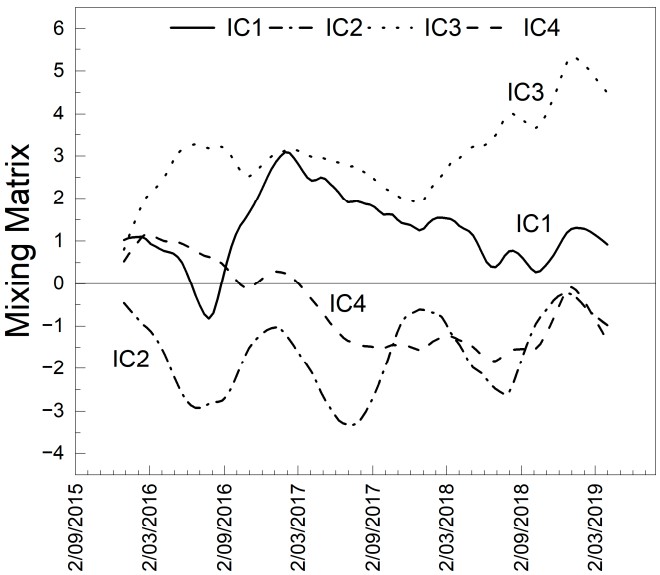

**Figure 10.** Temporal functions of the components extracted from T-ICA on the PS-DSs of the TerraSAR-X dataset that are located within the 10 m buffers around the cavity polygons. The smoothed lines are obtained from the results using the lowess method with 0.1 span.

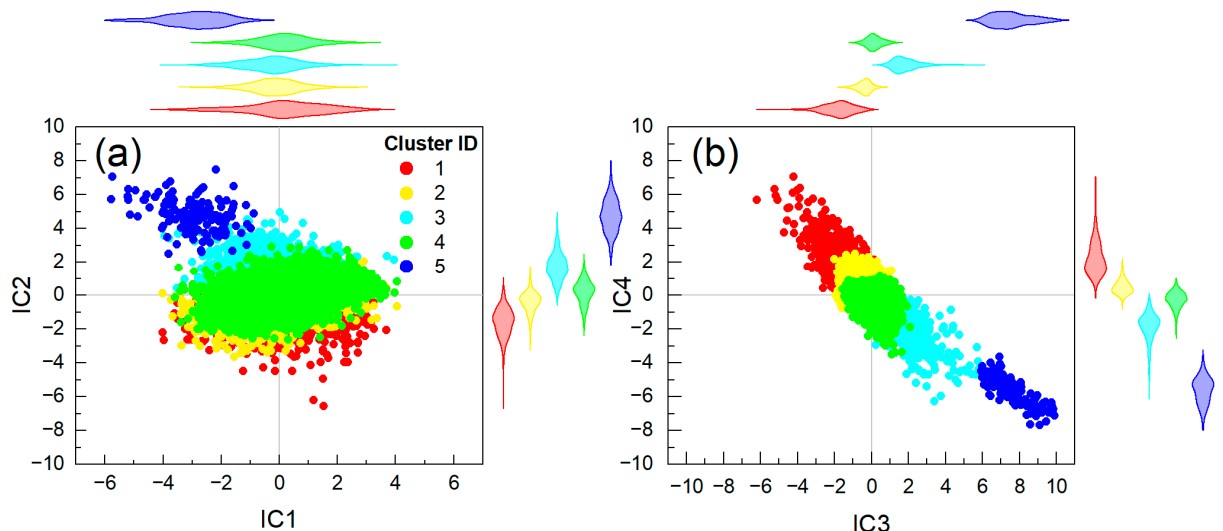

**Figure 11.** Five clusters obtained from T-ICA applied to the TerraSAR-X PS-DSs above the 10 m buffers of the cavity's polygons: (**a**) IC1-IC2 plane projection and (**b**) IC3-IC4 plane projection.

PS-DSs within each buffer cavity may belong to different clusters, which implies that the classification of cavities based only on clusters is insufficient to classify each cavity. For this purpose, the standard deviation of the cluster IDs for the PS-DSs within each buffer was computed (Figure 12a). Once the standard deviation threshold value of 1 was established, the cluster-ID mode (i.e., the most predominant cluster ID for each cavity) was calculated for each cavity within the threshold value, and that cluster ID was assigned to each cavity (Figure 12b). In contrast, cavities with a standard deviation greater than the threshold value were excluded from the classification because it was not possible to classify them due to the strong variability in the PS-DSs characterizing these cavities. Based on the results, among the 580 classified cavities, 12 were assigned to cluster ID 1, 195 to cluster ID 2, 57 to cluster ID 3, 305 to cluster ID 4 and 11 to cluster ID 5, as shown in Figure 12b.

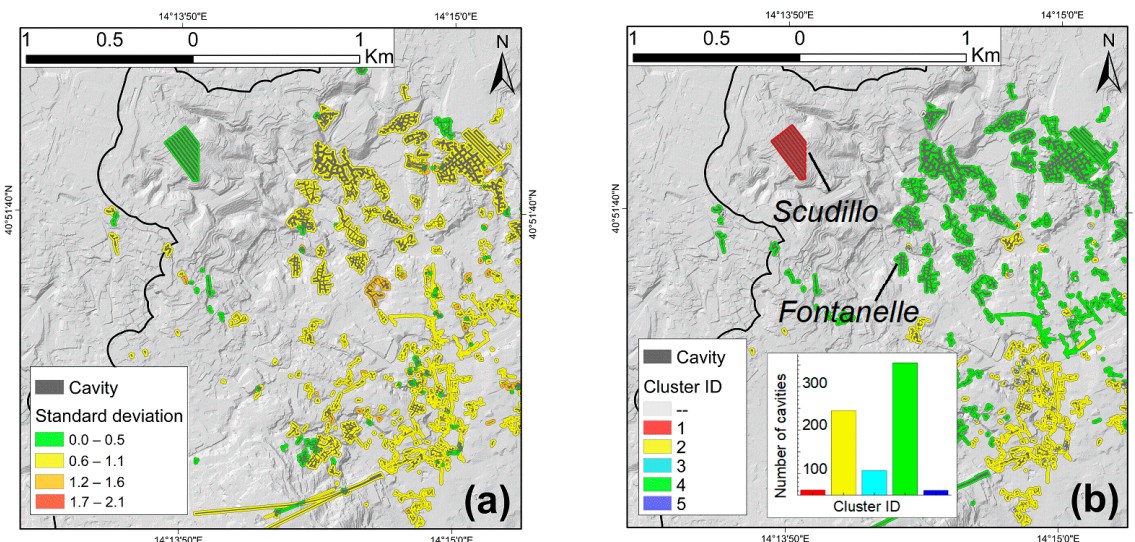

**Figure 12.** Maps of (**a**) the standard deviation of 10 m buffers around the cavity polygons; (**b**) the cavity classification based on the cluster ID mode within 10 m buffers around each cavity polygons and bar plots representing the number of cavities classified based on the cluster ID mode.

## 4. Discussion

Ref. [11] studied the effects of centering and standardization of the T- and S-PCA input matrix on temperature anomalies. They concluded that the first components extracted from the spatial mode represent the trend characterizing the largest amount (i.e., most of the analyzed area) of measurements points. In contrast, for the temporal mode, the results showed the spatial distribution of displacement rates of the considered component. These differences were initially noted by [57,58].

To better comprehend the effect due to the S-mode and T-mode matrix configuration, we performed the analyses on 20,000 PS-DSs that were spatially randomly extracted from the TerraSAR-X dataset taken from the northern sector of the UNESCO area to avoid uplift. As a result, the subsidence phenomenon is detected by components with an increasing or decreasing trend, depending on the positive or negative score signs. The temporal function of the first component describes a decreasing annual trend (Figure 13a), and the corresponding scores are reported in Figure 13c, where it is observed that the largest number of scores is positive. This means that the first component extracted from the S-PCA clearly describes the ground deformation that characterized most of the PS-DSs considered. Figure 13b,d show the results obtained from the T-PCA. The trend of PC1 has an increasing monotonic trend and the associated score map reveals positive and negatives values. This does not necessarily imply a different displacement mechanism because these values must be added to the mean displacement time series. Instead, in this case, they indicate different subsidence rates with respect to the average of the time series according to their score signs.

Therefore, our results confirm what was observed by [11,57] and the use of T- and S-modes for interpreting PS-DS displacements.

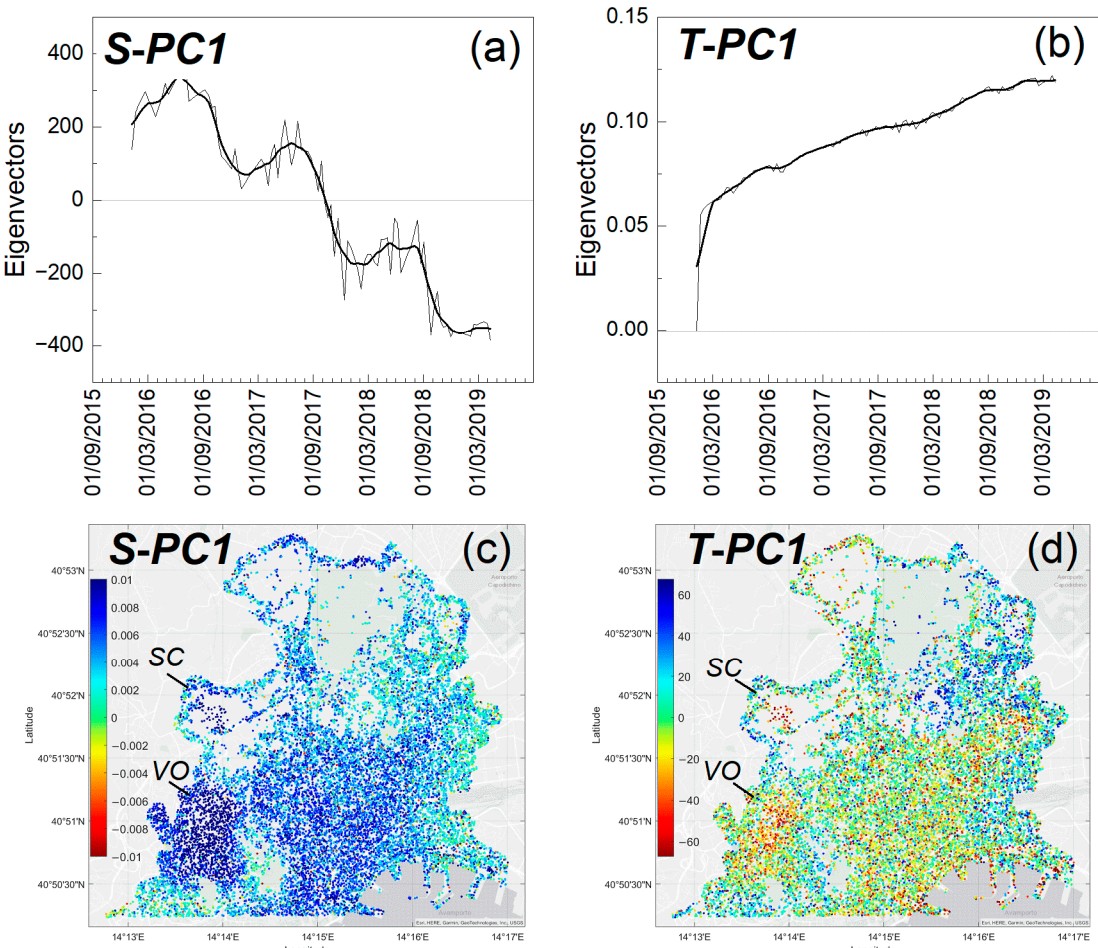

**Figure 13.** PCA of 20,000 PS-DSs extracted from TerraSAR-X dataset and covering the northern area of the UNESCO site. (**a**) Temporal function time series of S-PC1, (**b**) temporal function time series of T-PC1; (**c**) scores map of S-PC1 and (**d**) scores map of T-PC1. Black lines in (**a**,**b**) the smoothed lines obtained from the results using the lowess method with 0.1 span.

As stated earlier, the use of S-PCA and T-PCA allows us to classify the analyzed area into subregions based on the observed ground deformation trends. This classification was performed by [37], although they only considered velocities obtained from the ERS and Radarsat datasets, monitoring the period from 1992 to 2007. Comparing their findings with ours, the identified subregions coincide, but in addition, in the present work, it was possible to distinguish uplift and subsidence trends from seasonal trends.

As the subsurface of Naples is characterized by a widespread network of cavities (Figure 1), the stability assessment of underground cavities is a challenge that needs to be addressed. Indeed, these are often excavated at shallow depths relative to the ground level, thus affecting the stability of overlying structures and infrastructures. In addition, soft porous rocks, such as NYT, are generally very sensitive to weathering processes generated by moisture, water infiltration, saturation and fluid circulation [63]. Therefore, several methods have been presented in the literature to assess the stability of underground cavities. Generally, at the preliminary stage of analysis, phenomenological and analytical approaches are proposed to assess the stability condition. Some authors have proposed charts summarizing the general parameters to preliminarily assess the stability of the cavity obtained from numerical analysis [64]. The above methods are based on the equilibrium conditions, and therefore they analyzed the limit state conditions just before the failure

occurs. Consequently, their stability is based on the ratio between the failure condition and the current condition, but in contrast, our approach using the deformation trend is based on a kinematic condition. Nevertheless, the two approaches can give congruent results. For instance, Ref. [65] performed stability analyses for the Fontanelle cavity in Naples, and their results are in accordance with ours (Figure 12b).

Other authors, such as [66], considered velocities obtained from InSAR data to supplement useful information for susceptibility and hazard analyses, possibly allowing early prevention activities, especially when they are combined with in situ knowledge. In their works, the velocity takes into account all the phenomena taking place within their cavities, whereas our approach has the advantage of being able to classify the cavities separately considering the phenomena occurring simultaneously. Indeed, our classification, based on ICA components and clustering, has a more physical meaning since it is driven by the predominant mechanism.

In the cavity classification based on Hierarchical Clustering described in the results section, we considered a buffer size (see Figure 14a for the definition of the buffer size) surrounding each cavity equal to 10 m and performed the analysis considering PS-DSs located within the buffers. A radius of influence should be considered to estimate the areal extent potentially affected by subsidence related to a cavity in the subsurface. Ref. [24] proposed an analytical equation that defines the potential radius of the influence area around each cavity, where R is the radius of the influence, H is the cavity depth, and $\beta$ is the draw angle (see Figure 14b). By definition, the draw angle is the angle at which subsidence extends from the edge of the cavity towards the subsidence limit at the surface (see Figure 14b). The radius of the average influence area associated with the analyzed cavities was therefore computed by considering in the [24]'s equation an average cavity depth of 15 m [62] and a draw angle of 35° based on literature values for similar material [24]. An average influence buffer size of approximately 10 m was then considered around each cavity.

In the metropolitan area of Naples, the presence of unconsolidated pyroclastic deposits associated with both hard and weak volcanic rocks is widespread [32,67,68], and most of the cavities were excavated in NYT and the overlying layer is less than 20 m thick [44], as shown in Figure S7. As it is typically observed in the area, it was assumed that the tuff is generally found below an average 10 m thick layer of cohesionless soil called pozzolana [65]. In the kinematic classification of the cavities, we initially accounted for the average cavity depth and the typical $\beta$, derived from literature values. Subsequently, since $\beta$ is a function of the friction angle of the material, to account for the variability of ground parameters, we also considered the variation of the friction angle of the NYT and pozzolana and the depth range at which the cavities are located.

Starting from the friction angle parameter $\varphi$, which is 29° for the NYT and 33° for the pozzolana, we computed the values of $R$ (see Equations (3) and (4)), considering the depth range 5–35 m at which the cavities are excavated. According to Equations (3) and (4) we derived the values of $R$ considering all possible combinations of $\varphi$ and $H$ (see Figure 14b). The obtained $R$ values ranges from 2.9 m to 20.6 m. These results are agreement with that obtained in our study in which we observed that increasing the buffer size over 20 m the subsidence decreases and the data dispersion increases.

$$\alpha = \frac{\varphi}{2} + 45° \tag{3}$$

$$R = H \, \tan(90 - \alpha) \tag{4}$$

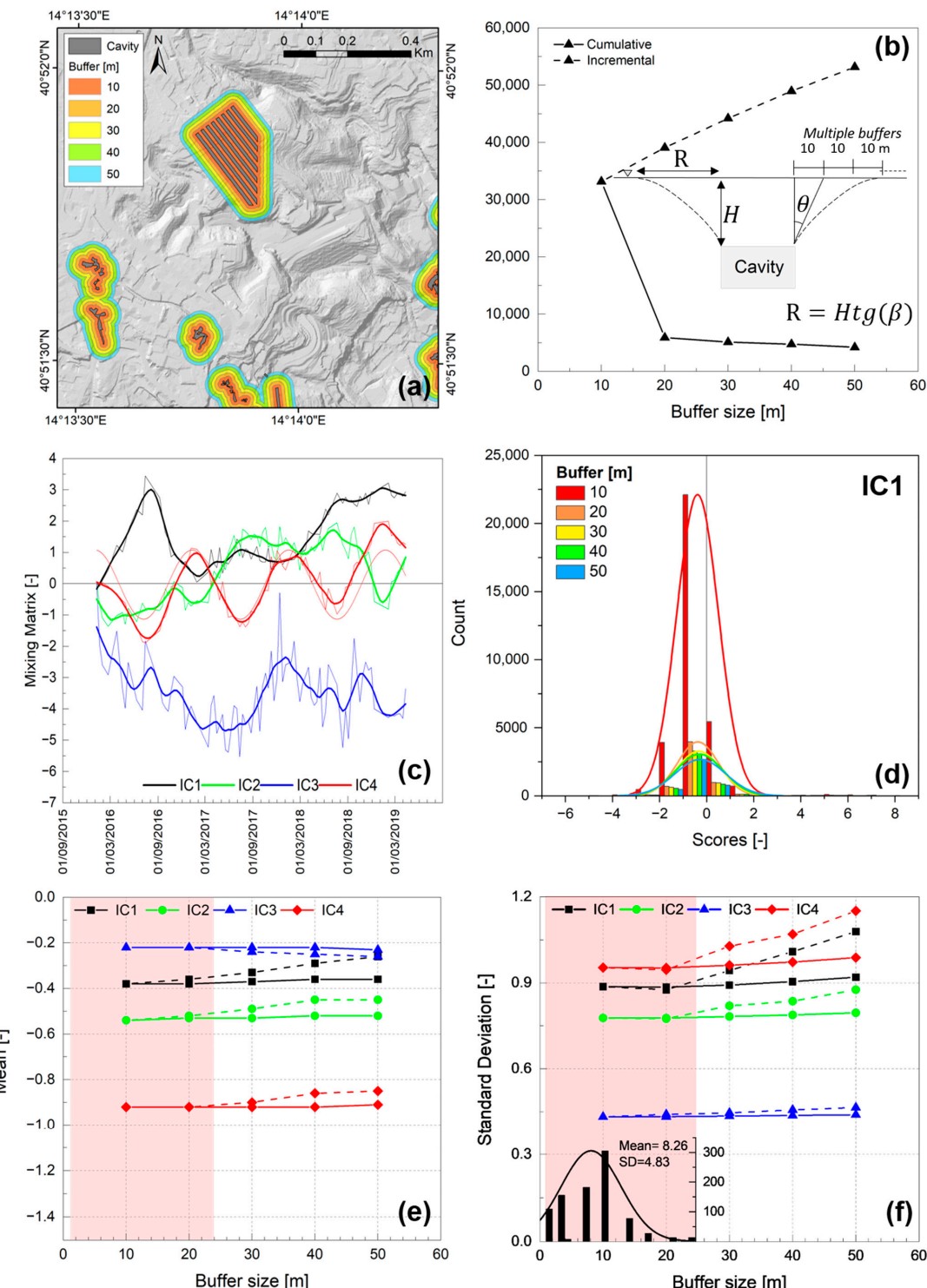

**Figure 14.** Results obtained considering different buffer sizes: (**a**) buffer representation; (**b**) number of PS-DSs as a function of the buffer size and representation of multiple buffers and draw angle, *β*, based on the equation proposed by [24]; (**c**) temporal function of T-IC components. The smoothed lines are obtained from the results using the lowess method with 0.1 span; (**d**) frequency distribution of the IC1 values within the multiple buffers; (**e**) mean values of the ICs as a function of buffer size; (**f**) standard deviation of the ICs as a function of buffer size with the frequency distribution of the influence radius. Shaded areas in (**e**,**f**) represent the ranges of the influence radius obtained from the maximum and minimum cavity depths using a *β* of 35°.

To investigate the effects of the buffer size, we performed a local ICA (T-mode) on the entire cavities inventory considering the PS-DSs located above the cavities and those within a buffer size of 50 m. The 50 m buffer was divided into further buffers with a step size of 10 m (see Figure 14a for the definition of the buffer size). The results are shown in Figure 14 where 'cumulative' represents the results obtained considering all PS-DS within the buffer considered, while 'incremental' represents the results obtained considering only the PS-DSs belonging to the rings between the buffer considered and the smallest nearby one. Figure 14b shows the number of PS-DSs belonging to each buffer, in which a decreasing trend is observed as we merged two datasets before the analysis; in one, we used the PS-DSs located within the buffer of size 10 m and in the other, the PS-DSs located within the 50 m buffer. The PS-DS dataset referring to the 50 m buffer was spatially randomly subsampled to 20,000 data points to reduce the computational burden. This procedure was adopted, despite the size of the buffer, to extract the effects of the presence of the cavity on the surrounding area. The evolution of the four extracted ICA components is shown in Figure 14c as a function of time. For all components, a large number of scores are negative and, therefore, the first and second components represent a trend related to subsidence, whereas the third component, which has the opposite sign, represents uplift, perhaps induced by the bradyseism phenomenon. Finally, the fourth oscillating component refers to a seasonal increasing trend. This signal can be fitted by a sinusoidal function of period $185 \pm 2$ days with peaks in winter and it may be caused by thermal effects.

If we observe the frequency distributions of the scores (as performed for Figure 14d), we obtain a Gaussian distribution characterized by the mean and a standard deviation. Thus, the effects of the buffer size can be quantified through the evolution of the mean and standard deviation values for each buffer. In particular, Figure 14e shows the variation in the mean value of the scores as a function of the buffer size for the different ICs. The obtained values, except for IC3, have an increasing trend with the buffer size, although the number of PS-DSs decreases with it. This implies that PS-DSs located outside the 10 m buffer size have very different mean values, which can influence the cumulative mean value despite their small number. Finally, for IC3, no great variations are observed in both cumulative and incremental terms, as the uplift affects the whole region of Naples. Analogous considerations can be given for the standard deviation, as reported in Figure 14f. It is necessary to consider that the study shows a limitation related to the estimation of vertical displacements by LOS projection along the vertical direction, as not from a combination of ascending and descending datasets, but only using descending datasets. Horizontal displacements broaden the area of influence of any cavity or stress/strain variation at depth.

Furthermore, since among the first extracted components, the linear trend and seasonal signal are always found, another possible approach could be to subtract these two components from the time series before performing PCA and ICA in order to be able to identify over time complex non-linear phenomena.

Finally, the elevations of PS-DSs were not considered in this work, which should instead be considered in order to be able to distinguish street-level PS-DSs from those located on the roofs of buildings, as their mix could introduce biases into the analyses.

## 5. Conclusions

In this paper, a unified algorithm was proposed that combines PCA, ICA and HC, considering both temporal and spatial modes to analyze InSAR displacement data. For this purpose, the algorithm was implemented in a self-built code and applied to the InSAR ERS1-2, Envisat, COSMO-SkyMed and TerraSAR-X datasets for the UNESCO site of Naples from 1992 to 2019. We confirm that both PCA and ICA allow the identification of sub-regions characterized by subsidence, uplift and/or seasonal ground deformations, decomposing the original time series into a set of extracted components. These components could be further associated with specific factors or controlling variables. ICA often allows a better separation of the different source signals. We prove that S-PCA gives among the

first components those with trends that are common to a broader number of PS-DSs (i.e., cover larger areas), while T-PCA highlights patterns with different ground deformation rates. Finally, clustering is an effective tool to group PS-DSs on the basis of these extracted components. The results support the use of this approach to classify cavities using local clustering. We compared the results obtained by analyzing PS-DS data from within different buffer areas around known underground cavities. We demonstrated that by increasing the buffer size, the effect of a cavity, in terms of subsidence, reduces. This agrees with the results obtained through the geomechanical-based draw angle method that allows us to define the expected extent of the affected area considering material properties and cavity depth.

In this work, we considered the available data, which include only LOS data acquired from satellites in descending geometry and obtained using PSI and SqueeSAR techniques. As a consequence, we did not compute the east–west displacement component by combining descending and ascending datasets. This could be a limitation; however, we mainly focused on local uplift, subsidence and sinkhole phenomena, where the main component of the displacement is the vertical one. These limitations of the methodology should be considered, and further research is needed to improve its accuracy and applicability to different scenarios. Anyway, this approach can be applied to E-W datasets, as well as to any other spatially distributed time series. One of the main advantages of the proposed methodology is the simple and standard algorithm and the speed of processing, even with large amounts of data. When combined with real–time monitoring data, the approach could be used to isolate areas characterized by changes in the trend of the time series and to forecast possible failure or crisis events. Future efforts will be focused on the use of this approach on pre-treated time series by eliminating linear and seasonal components.

**Supplementary Materials:** The following supporting information can be downloaded at: https://www.mdpi.com/article/10.3390/rs15123082/s1, Figure S1: Mean LOS velocity map from the TerraSAR-X dataset; Figure S2: (a) Comparison of CSK and cGPS MAFE measurements; (b–d) comparisons of TerraSAR-X with cGPS MAFE, cGPS ISMO and cGPS BAGN, respectively, in which the time intervals refer to the datasets. In every figure: grey shadow areas show the cGPS error measurements; red lines represent the average vertical displacements of PS-DSs at a buffer of 50 m from the cGPS stations; blue lines represent vertical displacements of the PSs closest to each cGPS station; Figure S3: Spatial distributions of the components extracted from S-PCA and S-ICA on 20,000\ randomly selected PS-DSs from TerraSAR-X within the UNESCO site. (a) S-PC1, (b) S-PC3, (c) S-IC2 and, (d) S-IC4; Figure S4: Spatial distributions of the components extracted from T-PCA on 20,000 randomly selected PS-DSs from TerraSAR-X within the UNESCO site. (a) T-PC1, (b) T-PC2, (c) T-PC3 and, (d) T-PC4; Figure S5: Spatial distributions of the components extracted from the T-ICA applied to 20,000 randomly selected PS-DSs from TerraSAR-X within the UNESCO site: (a) T-IC1, (b) T-IC2, (c) T-IC3 and, (d) T-IC4; Figure S6: (a) IC3-IC4 plane projection of the five clusters obtained from T-ICA applied to TerraSAR-X PS-DSs above the 10\ m buffers of the cavity's polygons. (b) Reconstructed time series of the five cluster centroids based on the four considered independent components; Figure S7: Spatial distribution of cavities and sinkholes inventories together with the NYT depth representation from the ground level. Different symbols are used for sinkholes located within 20\ m and between 20\ m and 30\ m from the cavities; Table S1: Table reporting the parameters characterizing five clusters obtained from the HC on the T-ICA analysis performed on the PS-DSs within the 20 m buffer around each cavity. The table also reports the parameters characterizing the ellipse related to each cluster. References [26,42,69] are cited in the supplementary materials.

**Author Contributions:** Conceptualization, S.R., G.D., P.F. and G.B.C.; Methodology, S.R., G.D., P.F. and G.B.C.; Software, S.R., G.D., P.F. and G.B.C.; Validation, G.D., P.F. and G.B.C.; Formal analysis, S.R., G.D., P.F. and G.B.C.; Investigation, S.R., G.D., P.F. and G.B.C.; Resources, P.F. and G.B.C.; Data curation, S.R., G.D., P.F. and G.B.C.; Writing—original draft, S.R. and G.D.; Writing—review & editing, G.D., P.F. and G.B.C.; Visualization, G.D., P.F. and G.B.C.; Supervision, G.D., P.F. and G.B.C.; Project administration, P.F. and G.B.C. All authors have read and agreed to the published version of the manuscript.

**Funding:** This research was funded by Ministero dell'Università e della Ricerca, grant number 2017HPJLPW.

**Data Availability Statement:** Not applicable.

**Acknowledgments:** We are grateful to Paolo M. Guarino (ISPRA) for providing the cavity database, to Tele-Rilevamento Europa (TRE) Altamira for the TerraSAR-X descending data of the study area, to Istituto Nazionale di Geofisica e Vulcanologia-Osservatorio Vesuviano (INGV-OV) for providing the cGPS data, and to A. Ferretti for his extremely valuable suggestions and encouragement. We would also like to acknowledge the 'URGENT' project (PRIN, Ministry of University and Research MUR) and the Progetto Dipartimenti di Eccellenza (Ministry of University and Research MUR) for partially funding this research.

**Conflicts of Interest:** The authors declare no conflict of interest.

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
