# Peer review of "A Multivariate Time Series Analysis of Ground Deformation Using Persistent Scatterer Interferometry"

_remotesensing, doi:10.3390/rs15123082_

Round 1

Reviewer 1 Report

The manuscript looks interesting but have several aspects which would need to be clarified and improved before it can be accepted for publication:

1) The authors should explain why they consider PSI technique, why they consider only descending LOS data, why they consider horizontal displacements are negligible and the posible effects introduced by these assumptions in the obtained results. Also the implications of the results on the CF and Naples area volcanic research and monitoring, as well as the general aplicability of the methods and results, if any.

2) Some other comments described by sections an line’s numbers are the following.

2.1) General. The format used for citations along the text is not the used by the journal. It must be corrected. Please see the instructions for authors.

2.2) Abstract, line 12. Please describe the meaning of abreviations the first time they appear.

2.3) Introduction. Please explain here the answer to comment 1), why authors do not consider results obtained using other InSAR processing techniques as SBAS or MSBAS,…. What is the precision obtained in the considered results.

Lines 143-163. Please relate this text with Figure 1, e.g. including the names of the areas cited in the text in the figure. Maybe it could be useful to include a new figure with the stratigraphy.

Lines 190-197. Please include here the reason why authors only consider descending data. 

Line 202. Why authors consider that horizontal displacement are negigible? It is a very strong assumption and should be justified. Is it done only for some areas or for all the study area. Considering, e.g., the results obtained by Pepe et al. (2019) and Castaldo et al. (2021) or in a different ambient by Fernández et al. (2018) this considerations should be done in a very well justified way, evaluating the posible impact on the results.

Fernández, J., J. F. Prieto, J. Escayo, A. G. Camacho, F. Luzón, K. F. Tiampo, M. Palano, T. Abajo, E. Pérez, J. Velasco, T. Herrero, G. Bru, I. Molina, J. López, G. Rodríguez-Velasco, I. Gómez, J. J. Mallorquí, 2018. Modeling the two- and three-dimensional displacement field in Lorca, Spain, subsidence and the global implications. Scientific Reports, 8:14782, https://www.nature.com/articles/s41598-018-33128-0.

Pepe, S., De Siena, L., Barone, A., ...Bianco, F., Tizzani, P., 2019. Volcanic structures investigation through SAR and seismic interferometric methods: The 2011–2013 Campi Flegrei unrest episode. Remote Sensing of Environment, 2019, 234, 111440

Castaldo, R., Tizzani, P., Solaro, G., 2021. Inflating Source Imaging and Stress/Strain Field Analysis at Campi Flegrei Caldera: The 2009–2013 Unrest Episode. Remote Sens. 2021, 13(12), 2298; https://doi.org/10.3390/rs13122298

2.4) Table 1 should be moved to page 5 or 6 afters its first citation.

2.5) Figure 5 should be located after line 392.

2.6) I consider the order of Figures 8 and 9 should be inverse.

2.7) Conclusions. Please describe here a summary of the answers to comments and suggestions in 1).

Author Response

The manuscript looks interesting but have several aspects which would need to be clarified and improved before it can be accepted for publication:

  • The authors should explain:

Why they consider PSI technique

We considered PSI and SqueeSAR techniques because our available datasets were processed by TRA-Altamira using these methodologies and our aim was to improve and test PCA and ICA approaches.

Why they consider only descending LOS data

We only considered the descending LOS data because unfortunately the available datasets were acquired from satellites in descending geometry. However, we are well aware that when considering only the descending data, the vertical displacement is obtained by projecting only the LOS displacement along the vertical direction, dividing it by the cosine of the incidence angle , and this is only possible method for the available. It would certainly be better to have both ascending and descending data and calculate the vertical and East-West displacement components by combining the two data sets. As further exercise we analysed the U-D and E-W components as available from the EGMS Copernicus project

Why they consider horizontal displacements are negligible

As above said, we only considered the vertical component of the displacement as a consequence of the available dataset. As only the descending datasets were available, it was not possible to compute the East-West displacement component. However, since the aim of this work is the characterization and classification of the vertical displacement mechanisms, we mainly focused on uplift, subsidence and sinkholes, where the main component of displacement is the vertical one. In particular, these mechanisms were revealed at the site scale (i.e., the whole area considered, the UNESCO World Heritage site of Naples) and the horizontal displacement is not relevant because, at the site conditions, the horizontal displacement component is related to local landslides, which are smaller than the site dimension. For the local analyses, we focused on cavities and sinkholes, where the horizontal displacement is not the main displacement component even if it can be useful for further analyses. A case apart is the Phlegrean Fields area, where the bradyseism phenomena cause both vertical and horizontal displacement components. Consequently, this mechanism is successfully revealed in the vertical displacement analysis and the horizontal displacement analysis is unnecessary.

And the possible effects introduced by these assumptions in the obtained results.

This is an important question. The authors have the opinion that one possible drawback consists in neglecting other mechanisms related to horizontal displacements or with an important horizontal component. However, the proposed methodology can also be applied to East-West displacement time series datasets.

Also, the implications of the results on the CF and Naples area volcanic research and monitoring, as well as the general applicability of the methods and results, if any.

This approach allows a large amount of data to be analysed efficiently: the approach is standardized and easily implemented in a numerical code. The results allow the identification of the main deformation trends decomposing the observed displacement time series in their source signals. Clustering provides a representation of the main spatial-temporal patterns involved in the area. For this reason, this approach can be combined with real-time monitored data for forecasting possible failure or crisis implying a sharp change in the behaviour.

2) Some other comments described by sections an line’s numbers are the following.

2.1) General. The format used for citations along the text is not the used by the journal. It must be corrected. Please see the instructions for authors.

Thanks to the reviewer, we corrected the citation according to the journal standards.

2.2) Abstract, line 12. Please describe the meaning of abbreviations the first time they appear.

Thanks to the reviewer, we added the meaning of the abbreviations in the manuscript when they were first used.

2.3) Introduction. Please explain here the answer to comment 1), why authors do not consider results obtained using other InSAR processing techniques as SBAS or MSBAS? What is the precision obtained in the considered results?

We are aware that over the last two decades, several InSAR processing techniques have been developed, most of which belong to one of two families of algorithms: Persistent Scatterers Interferometry SAR (PSInSAR™; (Ferretti et al., 2001)) and Small BAseline Subset (SBAS; (Berardino et al., 2002)). In particular, multi-interferogram techniques exploit a stack of several SAR images. SqueeSAR™ (Ferretti et al., 2011), which is an advanced multi-interferogram technique, and combines elements of both PSI and SBAS techniques, achieving a high density of measurement points over regions where distributed scatterers (DS) (identified by SBAS-type algorithms) are present, while at the same time conserving the information given by the permanent scatterers (PS) employed in PSI.

However, our data included only InSAR datasets that had already been processed with PSInSAR™ (Ferretti et al., 2001) and SqueeSAR™ techniques (Ferretti et al., 2011) by TRE ALTAMIRA; therefore, these types of data were available and we did not have available datasets obtained using the SBAS technique for comparisons. The analysis of raw satellite data, nevertheless, could be of great value to compare the potential of the different InSAR processing techniques could be applied to any dataset mode of distribute time series.

We explained the precision obtained in the considered results by implementing the following part of the updated manuscript at lines 204-216:

“The most influential factors affecting the quality of the measurements and therefore of the results obtained are the spatial density of the measurement points, the quality of radar targets, the atmospheric conditions at the acquisition time, the distance between the measurements point and the reference point (REF) and the number and temporal distribution of acquisition. Regarding the precision obtained by PSInSAR™ technique [1], in terms of differential displacement measurements is and considering the mean velocity is  [1]. For measurements obtained using the SqueeSAR™ technique [46], precision is defined by considering a dataset of at least  scenes covering a two-year period, for a measurement point located less than 1km from the REF. Precision is expressed in terms of standard deviation, which refers to the average displacement rate relative to the REF. The typical accuracy obtained from SqueeSAR™ analysis is lower than  for average annual velocity, while the single measurement is generally within  [47].”

Lines 143-163. Please relate this text with Figure 1, e.g. including the names of the areas cited in the text in the figure. Maybe it could be useful to include a new figure with the stratigraphy.

We modified Figure 1, introducing the location of the named areas and the sinkholes distribution. The stratigraphy related to the Scudillo area refers to (Terranova et al., 2015).

Lines 190-197. Please include here the reason why authors only consider descending data.

As explained before, our available datasets include only descending data.

Line 202. Why authors consider that horizontal displacement are negigible? It is a very strong assumption and should be justified. Is it done only for some areas or for all the study area. Considering, e.g., the results obtained by Samsonov et al. (2014), Pepe et al. (2019) and Castaldo et al. (2021) or in a different ambient by Fernández et al. (2018) this considerations should be done in a very well justified way, evaluating the posible impact on the results.

Samsonov, S. V., K. F. Tiampo, A. G. Camacho, J. Fernández, and P. J. González, 2014. Spatiotemporal analysis and interpretation of 1993–2013 ground deformation at Campi Flegrei, Italy, observed by advanced DInSAR. Geophysical Research Letters, 41, 6101-6108, doi: 10.1002/2014GL060595.

Fernández, J., J. F. Prieto, J. Escayo, A. G. Camacho, F. Luzón, K. F. Tiampo, M. Palano, T. Abajo, E. Pérez, J. Velasco, T. Herrero, G. Bru, I. Molina, J. López, G. Rodríguez-Velasco, I. Gómez, J. J. Mallorquí, 2018. Modeling the two- and three-dimensional displacement field in Lorca, Spain, subsidence and the global implications. Scientific Reports, 8:14782, https://www.nature.com/articles/s41598-018-33128-0.

Pepe, S., De Siena, L., Barone, A., ...Bianco, F., Tizzani, P., 2019. Volcanic structures investigation through SAR and seismic interferometric methods: The 2011–2013 Campi Flegrei unrest episode. Remote Sensing of Environment, 2019, 234, 111440

Castaldo, R., Tizzani, P., Solaro, G., 2021. Inflating Source Imaging and Stress/Strain Field Analysis at Campi Flegrei Caldera: The 2009–2013 Unrest Episode. Remote Sens. 2021, 13(12), 2298; https://doi.org/10.3390/rs13122298

The reviewer is correct, we had no data on the horizontal displacement since a combination of ascending and descending datasets is required to obtain this information. Therefore we focused on the identification and classification of only the vertical behavior of displacements in the UNESCO World Heritage site of Naples (i.e. the prevailing vertical component associated with subsidence/uplift, sinkholes and cavity instability, generally of small size), so we did not consider horizontal displacements in our analysis. The velocities and horizontal displacements cannot be derived from the available datasets, as all datasets were acquired from satellites in descending orbit. Nevertheless, we are well aware of the relevance of horizontal displacements in subsidence/uplift phenomena, in particular when analyzing the Phlegrean Fields ground deformation mechanisms. Therefore, we modified the text in the updated manuscript at lines 238-248:

 “As previously mentioned, InSAR displacements are measured along the satellite LOS. Since the aim of this study is to describe and classify vertical displacement mechanisms, the measured velocities and displacements of each InSAR dataset were projected along the vertical direction. In this work, we only considered the vertical component of the displacement as a consequence of the available dataset. Since only the descending dataset was available, it was not possible to calculate the East-West component of the displacement. However, we mainly focused on uplift, subsidence and sinkholes phenomena, for which the main component of the displacement is the vertical one. To achieve this, the velocities and displacements along the LOS were divided by the cosine of the local incidence angle  (i.e., the angle between the LOS and the vertical direction - see values in Table1). “

2.4) Table 1 should be moved to page 5 or 6 afters its first citation.

[Line 249] Thanks to the reviewer, we moved Table 1 to page 6.

2.5) Figure 5 should be located after line 392.

[Line 442] Thanks to the reviewer, we moved Figure 5 after line 441.

2.6) I consider the order of Figures 8 and 9 should be inverse.

[Lines 543 and 545] Thanks to the reviewer, we moved Figure 8 and Figure 9.

2.7) Conclusions. Please describe here a summary of the answers to comments and suggestions in 1).

Summarizing the answers to the comments and suggestions of the reviewer, we implemented the conclusions section by adding the following part of the updated manuscript at lines 754-769:

“In this work, we considered the available data, which includes only LOS data acquired from satellites in descending geometry and obtained using PSI and SqueeSAR techniques. As a consequence we did not compute the East-West displacement component by combining descending and ascending datasets. This could be a limitation, however, we mainly focused on local uplift, subsidence and sinkhole phenomena, where the main component of the displacement is the vertical one. These limitations of the methodology should be considered, and further research is needed to improve its accuracy and applicability to different scenarios. Anyway, this approach can be applied to E-W datasets as well as to any other spatially distributed time series. One of the main advantages of the proposed methodology is the simple and standard algorithm and the speed of processing, even with large amounts of data. When combined with real-time monitoring data, the approach could be used to isolate areas characterized by changes in the trend of the time series and to forecast possible failure or crisis events. Future efforts will be focused on the use of this approach on pre-treated time series by eliminating linear and seasonal components.”

Other manuscript implementations:

We also implemented the text with the following sentence related to Figure S 8 reported in the Supplementary Materials section at lines 564-567:

“The statistical and the 95% confidence ellipse parameters of each cluster are reported in Table S1. The centroid of each cluster, corresponding to a PS-DS, was identified and the original time series was compared with that reconstructed from the considered four ICs, as shown in Figure S 2.

T-ICA and clustering for cavity classification at the local scale: we rephrased in the updated manuscript at lines 571-576:

 “Once the standard deviation threshold value of 1 was established, the cluster-ID mode (i.e., the most predominant cluster ID for each cavity) was calculated for each cavity within the threshold value, and that cluster ID was assigned to each cavity (Figure12b). In contrast, cavities with a standard deviation greater than the threshold value were excluded from the classification because it was not possible to classify them due to the strong variability in the PS-DSs characterizing these cavities.”

Discussions: We implemented the discussion section by adding some considerations in the updated manuscript at lines 723-734:

 “It is necessary to consider that the study shows a limitation related to the estimation of vertical displacements by LOS projection along the vertical direction, as not from a combination of ascending and descending datasets, but only using descending datasets. Horizontal displacements broaden the area of influence of any cavity or stress/strain variation at depth.

Furthermore, since among the first extracted components, the linear trend and seasonal signal are always found, another possible approach could be to subtract these two components from the time series before performing PCA and ICA in order to be able to identify over time complex nonlinear phenomena.

Finally, the elevations of PS-DSs were not considered in this work, which should instead be considered in order to be able to distinguish street-level PS-DSs from those located on the roofs of buildings, as their mix could introduce biases into the analyses.”

Conclusions: we implemented the conclusions section by adding some considerations in lines 754-761:

“In this work, we considered the available data, which includes only LOS data acquired from satellites in descending geometry and obtained using PSI and SqueeSAR techniques. As a consequence we did not compute the East-West displacement component by combining descending and ascending datasets. This could be a limitation, however, we mainly focused on local uplift, subsidence and sinkhole phenomena, where the main component of the displacement is the vertical one. These limitations of the methodology should be considered, and further research is needed to improve its accuracy and applicability to different scenarios.”

Supplementary material:

Considering the sinkholes inventory, we implemented the Supplementary Materials section by adding the following considerations at lines 801-816:

 “Based on the recent inventory of “anthropogenic” sinkholes in the city of Naples [27], we compared the location of sinkholes with the cavity inventory [70-71]. Among the total sinkholes, twenty are located within a  buffer area surrounding each cavity. When increasing the buffer size from  to , twenty-two sinkholes were identified in the cavity surroundings (Figure S 8). This outcome suggests that sinkholes located at a distance greater than  from cavities could probably be not related to their presence. Furthermore, according to the results obtained by T-ICA performed on the PS-DSs located within the different buffer sizes (from  to , see Figure14b), increasing the distance from the cavity, the subsidence effect markedly decreases and the dispersion of the data increases (as shown in Figure14e-f), particularly starting from from the cavity.

Subsequently, we associated each selected sinkhole located within the buffer size of with the nearby cavity and considered only the eight sinkholes that occurred during the period monitored by TerraSAR-X ( ). Among them, it was observed that  are related to cavities classified as cluster ID 2, and therefore characterized by less pronounced subsidence kinematic.

Reviewer 2 Report

The Authors show the application of different statistical models applied to satellite data for the analysis of ground deformations in a portion of the city of Naples. The work seems very interesting but  I think that it has several shortcomings in different parts of the text. First of all, for the data used the authors probably did not consult the latest works published in international journals, also for what concerns the geological framework one should refer to surely more detailed works (Coda et al., 2019 is surely not the most appropriate). Also, for the SAR Interferometry part, the authors should have a broader view of the literature and not only refer to the algorithm used. In the description of the investigated area there is no mention of the Unesco area, which is instead depicted in Figure 1; it would be useful to specify from the outset that the analyses were carried out in this context. Another questionable aspect is the assumption made regarding the main component of deformation, which the authors identify as vertical. As is well known, the morphology of the territory analysed is rather complex, in particular for the Posillipo hill; therefore, to carry out the projection simply by dividing by the cosine of the angle of incidence seems to me to be incorrect. As far as the images are concerned, the type is not specified (Stripmap, Spotlight....), but reporting the spatial resolution of 1x1 m in Table 1 should suggest that Spotlight images were used.

Is this correct? Because otherwise the resolution should be 3x3 m (Stripmap images). The analyses identified on a regional scale by the authors would also appear to be insignificant, as they focus on an area that is not so extensive and therefore not representative of the actual deformation trend. What I consider most important, however, is that all the data that are cited in terms of displacement rates refer exclusively to satellite elaborations and never to in-situ data. Like, for example, the extrapolation made to classify the cavities, from what I understand, does not take into account any ground parameters, such as the thickness of the pyroclastic covers or tuff. It would appear that the authors have carried out all the analyses without any ground feedback, which, as is well known from the hundreds of papers in international journals, is crucial to the validation of the satellite data.

Other comments:

Coordinates must be given in all figures.

All acronyms used for the first time must be described (JADE, SOBI, FOTBI, T-PCA, FCF .....).

Pg. 2 lines 57-60: this would appear to be a repetition of lines 54-56. Please, rephrase.

Pg. 2 lines 77-80. For the sake of completeness, it might be useful to cite works that have addressed these issues in the city of Naples.

Pg. 3: Study area: completely revise bibliography

Pg. 4 lines 141-142: cite the source

Pg. 4 line 143: the statement "Sinkholes are mainly located in the hilly areas" is incorrect. This also affects what has already been noted regarding the calculation of the deformation component.

Pg. 4 lines 149-151: Cite the source of the recorded displacement rate values.

Pg. 5 line 206: could the linear interpolation performed influence the results?

Fig. 2c: the date is wrong (2012-2012)

Table 1: Check the resolution values for what has already been stated above (image type). Having calculated the density, it would be useful to also report the areal extensions.

Pg. 11 line 355: should be better specified, it is not clear how the analysis was carried out at regional scale. Which areal extensions were worked on?

Pg. 11 line 360-362: it would be useful to carry out a consistency analysis regarding the number of PS/DS extracted from the CSK and TSX databases. They would appear to be insignificant in relation to numerosity. Also, how was the random extraction conducted? With a specific algorithm?

Pg. 11 lines 380-386: some of the observations do not seem to be supported by Figure 5. In particular, the trend of IC4 does not seem to be temporal but rather to have a similar trend to PC1 (linear). In general, the interpretation of these trends does not seem to be related to any physical processes that may be present.

In Figure 1, insignificant elements are depicted, in particular all the linear elements it is not clear what they could be used for, they should be removed. The cavities are also not clearly visible and the area is shown in a different colour.

Therefore, I believe that the work needs to be thoroughly revised, responding accurately to all comments

Author Response

The Authors show the application of different statistical models applied to satellite data for the analysis of ground deformations in a portion of the city of Naples. The work seems very interesting but I think that it has several shortcomings in different parts of the text. First of all, for the data used the authors probably did not consult the latest works published in international journals also for what concerns the geological framework one should refer to surely more detailed works (Coda et al., 2019 is surely not the most appropriate. Also, for the SAR Interferometry part, the authors should have a broader view of the literature and not only refer to the algorithm used. In the description of the investigated area there is no mention of the Unesco area which is instead depicted in Figure 1; it would be useful to specify from the outset that the analyses were carried out in this context. Another questionable aspect is the assumption made regarding the main component of deformation, which the authors identify as vertical. As is well known, the morphology of the territory analysed is rather complex, in particular for the Posillipo hill; therefore, to carry out the projection simply by dividing by the cosine of the angle of incidence seems to me to be incorrect. As far as the images are concerned, the type is not specified (Stripmap, Spotlight....), but reporting the spatial resolution of 1x1 m in Table 1 should suggest that Spotlight images were used

Is this correct? Because otherwise the resolution should be 3x3 m (Stripmap images).

The InSAR data analysed, with regard to the COSMO-SkyMed and TerraSAR-X datasets, are the results of SAR image datasets formed by stacks of StripMap (SM) images. We added the image and resolution information by implementing the text of the updated manuscript at lines 226-237 as follows:

”The InSAR data analyzed, with regard to the COSMO-SkyMed and TerraSAR-X datasets, are the results of SAR image datasets formed by stacks of StripMap (SM) images. The SM is the basic SAR imaging mode. The ground swath is illuminated with continuous sequence of pulses while the antenna beam is fixed in elevation and azimuth. This produces an image strip with a continuous quality (in the direction of flight). SM data with double polarization, such as those from which the analyzed data are derived, have a slightly lower spatial resolution and a smaller swath than the single polarization data. In SM mode, a spatial resolution of up to  can be achieved. Twin polarisation SM data recorded in HH (single polarisation channel) have a standard scene size of  (width x length). Specifically, in this work the area of the UNESCO World Heritage site of Naples was analyzed, which spans a surface of  in total (Figure1).”

The analyses identified on a regional scale by the authors would also appear to be insignificant, as they focus on an area that is not so extensive and therefore not representative of the actual deformation trend.

The reviewer's comment regarding the size of the regional scale is pertinent. We changed the term "regional scale" to "site scale". With the term “site scale” we referred to the UNESCO World Heritage site scale, which extends over . However, we do not fully agree with the reviewer's opinion regarding the representativeness of the dataset. In particular, the TerraSAR-X dataset consists of 2,566,269 PS-DSs considering the entire dataset ( ), and 404,281 considering only the site area ( ), which is the one that was actually analysed. Although we subsampled the TSX dataset to 20,000 PS-DSs, to reduce the computational cost, this resampling was done in a spatially random manner and the results obtained with different subsampled data were the same. This means that the data used in our algorithm are representative of the deformation trends of the analysed area.

We corrected the sentence in the updated manuscript at lines 408-410 as follows:

 “Site scale analyses were performed at the UNESCO World Heritage site scale ( ), to identify the main temporal and spatial patterns of ground deformation, whereas local scale analyses were performed at the cavity scale to classify the ground deformation related to the presence of cavities.”

What I consider most important, however, is that all the data that are cited in terms of displacement rates refer exclusively to satellite elaborations and never to in-situ data. Like, for example, the extrapolation made to classify the cavities, from what I understand, does not take into account any ground parameters, such as the thickness of the pyroclastic covers or tuff. It would appear that the authors have carried out all the analyses without any ground feedback, which, as is well known from the hundreds of papers in international journals, is crucial to the validation of the satellite data.

Therefore, I believe that the work needs to be thoroughly revised, responding accurately to all comments

We appreciate this insightful and valuable comment; the idea is to find typical precursor movement trends, and verify similar behaviour by knowing the location and size of cavities, and comparing them with areas that are stable. Moreover, our classification methodology is based only on kinematic and not static aspects, characterising the cavities according to the predominant vertical displacement trend. In particular, we used, in the buffer size evaluation, the average values of the mechanical properties observed within the area. This was done because we characterised all cavities at the same time, and each cavity has its own geometric parameters (i.e, planimetry and depth) and geological profile. If only data measured in the vicinity of a single, specific cavity were used, PCA, ICA and HC analysis would be unreliable due to the small amount of data considered.

In order to take into account ground parameters, such as the thickness of the pyroclastic covers and tuff we implemented the following part in the updated manuscript at lines 663-682 :

 “In the metropolitan area of Naples, the presence of unconsolidated pyroclastic deposits associated with both hard and weak volcanic rocks is widespread [33, 68], and most of the cavities were excavated in NYT and the overlying layer is less than  thick [44], as shown in FigureS 8. As it is typically observed in the area, it was assumed that the tuff is generally found below an average  thick layer of cohesionless soil called pozzolana [66]. In the kinematic classification of the cavities, we initially accounted for the average cavity depth and the typical , derived from literature values. Subsequently, since  is a function of the friction angle of the material, to account for the variability of ground parameters, we also considered the variation of the friction angle of the NYT and pozzolana and the depth range at which the cavities are located. Starting from the friction angle parameter , which is for the NYT and  for the pozzolana, we computed the values of  (equation 3 and 4), considering the depth range  at which the cavities are excavated. According to equations (3) and (4) we derived the values of  considering all possible combinations of  and (see Figure14b). The obtained values ranges from  and . These results are agreement with that obtained in our study in which we observed that increasing the buffer size over  the subsidence decreases and the data dispersion increases.

                                                                                                   (3)

                                   (4)”

Regarding the validation of the satellite data we implemented the following part in the updated manuscript at lines 293-304:

“To validate the InSAR data used in this work, the satellite-based ground measurements were compared with the displacements recorded by cGPS stations related to the Neapolitan Volcanoes cGPS (NeVoCGPS) network, operated by the Istituto Nazionale di Geofisica e Vulcanologia–Osservatorio Vesuviano (INGV-OV). For this purpose, we used cGPS measurements acquired on a weekly basis between January  and March . We considered the cGPS stations MAFE and NAMM operating in the Phlegraean Fields area [46-48, shown in Figure1. To validate the InSAR data, we considered the vertical component time series of the two cGPS stations and compared them with the COSMO-SkyMed and TerraSAR-X time series, projected in the vertical direction as explained in this section. The comparisons between InSAR data and cGPS show a reasonable match between the two datasets (  error bands for the cGPS measurements are given in FigureS1 and FigureS2).”

Other comments:

Coordinates must be given in all figures.

Thanks to the reviewer, we added coordinated in each figure.

All acronyms used for the first time must be described (JADE, SOBI, FOTBI, T-PCA, FCF .....).

Thanks to the reviewer, we added in the manuscript the meaning of each acronyms.

Pg. 2 lines 57-60: this would appear to be a repetition of lines 54-56. Please, rephrase.

Thanks to reviewer. We modified the sentence adding the information that in the second phrase the PCA was combined with the clustering analysis.

Pg. 2 lines 77-80. For the sake of completeness, it might be useful to cite works that have addressed these issues in the city of Naples.

Thanks to reviewer we cited the works.

Pg. 3: Study area: completely revise bibliography

Thanks to the reviewer, we revised the “Geological and hydrogeological settings” and “InSAR datasets” in the study area section.

Pg. 4 lines 141-142: cite the source

Thanks to the reviewer, we cited the source.

Pg. 4 line 143: the statement "Sinkholes are mainly located in the hilly areas" is incorrect. This also affects what has already been noted regarding the calculation of the deformation component.

Thanks to the reviewer, we corrected the sentence.

Pg. 4 lines 149-151: Cite the source of the recorded displacement rate values.

Thanks to the reviewer, we cited the source of the recorded displacement rate values within the Scudillo water reservoir area. In fact, the values were obtained from the analysed TerraSAR-X dataset.

Pg. 5 line 206: could the linear interpolation performed influence the results?

The subsampling methodology by linear interpolation was carried out to improve the quality of PCA. However, this modification introduces some artefacts. To appreciate the effects of this modification, Festa et al. (2023) compared the results obtained considering three different interpolations methods showing that the linear interpolation is the one that best approximates the original time series.

Fig. 2c: the date is wrong (2012-2012)

Thanks to the reviewer, we corrected the date: 2012-2013.

Table 1: Check the resolution values for what has already been stated above (image type). Having calculated the density, it would be useful to also report the areal extensions.

Thanks to the reviewer, we corrected the resolution values and reported the areal extentions of each dataset in Table1.

Pg. 11 line 355: should be better specified, it is not clear how the analysis was carried out at regional scale. Which areal extensions were worked on?

Thanks to the reviewer, we specified that the site scale analyses were performed at the UNESCO World heritage site scale. The considered area covers . For this reason, we included in the updated manuscript the following part at lines 408-411:

“Site scale analyses were performed at the UNESCO World Heritage site scale ( ), to identify the main temporal and spatial patterns of ground deformation, whereas local scale analyses were performed at the cavity scale to classify the ground deformation related to the presence of cavities.”

Pg. 11 line 360-362: it would be useful to carry out a consistency analysis regarding the number of PS/DS extracted from the CSK and TSX databases. They would appear to be insignificant in relation to numerosity. Also, how was the random extraction conducted? With a specific algorithm?

Subsampling of the original data set was done for computational issues. The authors implemented an algorithm in which a subset is randomly extracted from the original dataset. To exclude any influence on the determination of the displacement components due to the subsampling, the authors ran different random subsets, containing different PS-DSs. The results obtained with these datasets are comparable, and therefore the authors concluded that the subsampling does not alter the results of the analyses.

The authors are aware that employing the entire dataset may provide more precise results in terms of displacement trends, but due to the limitations discussed above, it is not possible to accomplish these results.

Pg. 11 lines 380-386: some of the observations do not seem to be supported by Figure 5. In particular, the trend of IC4 does not seem to be temporal but rather to have a similar trend to PC1 (linear). In general, the interpretation of these trends does not seem to be related to any physical processes that may be present.

We do not fully understand the reviewer made his comment about the non-correspondence between the observation and Figure5. In fact, we stated exactly what the reviewer claimed in the question. Regarding the physical process, the reviewer is right because we proposed an analysis whose results are used for a kinematic classification of the displacement mechanisms. This implies that we did not associate the displacement mechanisms with the possible underlying physical processes that generate them, which would require datasets for other variables, and this contribution is beyond the scope of this work.

In Figure 1, insignificant elements are depicted, in particular all the linear elements it is not clear what they could be used for, they should be removed. The cavities are also not clearly visible and the area is shown in a different colour.

Thanks to the reviewer for the suggestions, we modified Figure1.

Other manuscript implementations:

We also implemented the text with the following sentence related to Figure S 8 reported in the Supplementary Materials section at lines 564-567:

“The statistical and the 95% confidence ellipse parameters of each cluster are reported in Table S1. The centroid of each cluster, corresponding to a PS-DS, was identified and the original time series was compared with that reconstructed from the considered four ICs, as shown in Figure S 2.

T-ICA and clustering for cavity classification at the local scale: we rephrased in the updated manuscript at lines 571-576:

 “Once the standard deviation threshold value of 1 was established, the cluster-ID mode (i.e., the most predominant cluster ID for each cavity) was calculated for each cavity within the threshold value, and that cluster ID was assigned to each cavity (Figure12b). In contrast, cavities with a standard deviation greater than the threshold value were excluded from the classification because it was not possible to classify them due to the strong variability in the PS-DSs characterizing these cavities.”

Discussions: We implemented the discussion section by adding some considerations in the updated manuscript at lines 723-734:

 “It is necessary to consider that the study shows a limitation related to the estimation of vertical displacements by LOS projection along the vertical direction, as not from a combination of ascending and descending datasets, but only using descending datasets. Horizontal displacements broaden the area of influence of any cavity or stress/strain variation at depth.

Furthermore, since among the first extracted components, the linear trend and seasonal signal are always found, another possible approach could be to subtract these two components from the time series before performing PCA and ICA in order to be able to identify over time complex nonlinear phenomena.

Finally, the elevations of PS-DSs were not considered in this work, which should instead be considered in order to be able to distinguish street-level PS-DSs from those located on the roofs of buildings, as their mix could introduce biases into the analyses.”

Conclusions: we implemented the conclusions section by adding some considerations in lines 754-761:

“In this work, we considered the available data, which includes only LOS data acquired from satellites in descending geometry and obtained using PSI and SqueeSAR techniques. As a consequence we did not compute the East-West displacement component by combining descending and ascending datasets. This could be a limitation, however, we mainly focused on local uplift, subsidence and sinkhole phenomena, where the main component of the displacement is the vertical one. These limitations of the methodology should be considered, and further research is needed to improve its accuracy and applicability to different scenarios.”

Supplementary material:

Considering the sinkholes inventory, we implemented the Supplementary Materials section by adding the following considerations at lines 801-816:

 “Based on the recent inventory of “anthropogenic” sinkholes in the city of Naples [27], we compared the location of sinkholes with the cavity inventory [70-71]. Among the total sinkholes, twenty are located within a  buffer area surrounding each cavity. When increasing the buffer size from  to , twenty-two sinkholes were identified in the cavity surroundings (Figure S 8). This outcome suggests that sinkholes located at a distance greater than  from cavities could probably be not related to their presence. Furthermore, according to the results obtained by T-ICA performed on the PS-DSs located within the different buffer sizes (from  to , see Figure14b), increasing the distance from the cavity, the subsidence effect markedly decreases and the dispersion of the data increases (as shown in Figure14e-f), particularly starting from from the cavity.

Subsequently, we associated each selected sinkhole located within the buffer size of with the nearby cavity and considered only the eight sinkholes that occurred during the period monitored by TerraSAR-X ( ). Among them, it was observed that  are related to cavities classified as cluster ID 2, and therefore characterized by less pronounced subsidence kinematic.

Round 2

Reviewer 1 Report

After checking the revised ms, I consider it can be accepted after some minor revision:   1) After line 248 I propose to the authors to include some text from their anser to my previous questions "Why they consider horizontal displacements are negligible" and "And the possible effects introduced by these assumptions in the obtained results".   2) Lines 254-255. The authors write that "All displacement time series, projected along the vertical direction, were then linearly interpolated to resample them with a regular acquisition in time." Is this interpolation necessary? why? please explain it and also if it have some effects on the results.   3) I have observed that, at least in my version, quality of Figures S1 and S2 is not good. The displacements shown in the stations are very small, few mm, and then clearly the differences are also small. The authos should select GNSS with greater dsplacemntes (at least several cm per year) to show clearly the differences, Please, also explain what is the meaning of the red lines in these figures in their captions

Reviewer 2 Report

I observed the authors' answers very carefully, and I could see a considerable improvement in the manuscript. Almost all the comments were answered comprehensively by the authors, in my opinion, only in a few cases the answers were not completely comprehensive (evaluation of the components in the geomorphological context, consistency of the extracted PS with respect to the dataset). In any case, I do not believe that the text can be further improved and therefore feel that it can be accepted in its current form.
